# Optical Coherence Tomography Harmonization with Anatomy-Guided Latent Metric Schrödinger Bridges

**Shuwen Wei**[1][*]   **Samuel W. Remedios**[1]   **Blake E. Dewey**[2]   **Zhangxing Bian**[1]

**Shimeng Wang**[1]   **Junyu Chen**[2]   **Bruno M. Jedynak**[3]   **Shiv Saidha**[2]

**Peter A. Calabresi**[2]   **Aaron Carass**[1]   **Jerry L. Prince**[1]

[1]Johns Hopkins University   [2]Johns Hopkins School of Medicine

[3]Portland State University

## Abstract

Medical image harmonization aims to reduce the differences in appearance caused by scanner hardware variations to allow for consistent and reliable comparisons across devices. Harmonization based on paired images from different devices has limited applicability in real-world clinical settings. On the other hand, unpaired harmonization typically does not guarantee anatomy consistency, which is problematic because anatomical information preservation is paramount. The Schrödinger bridge framework has achieved state-of-the-art style transfer performance with natural images by matching distributions of unpaired images, but this approach can also introduce anatomy changes when applied to medical images. We show that such changes occur because the Schrödinger bridge uses the square of the Euclidean distance between images as the transport cost in an entropy-regularized optimal transport problem. Such a transport cost is not appropriate for measuring anatomical distances, as medical images with the same anatomy need not have a small Euclidean distance between them. In this paper, we propose a latent metric Schrödinger bridge (LMSB) framework to improve the anatomical consistency for the harmonization of medical images. We develop an invertible network that maps medical images into a latent Euclidean metric space where the distances among images with the same anatomy are minimized using the pullback latent metric. Within this latent space, we train a Schrödinger bridge to match distributions. We show that the proposed LMSB is superior to the direct application of a Schrödinger bridge to harmonize optical coherence tomography (OCT) images.

## 1   Introduction

Optical coherence tomography (OCT) (Huang et al., 1991) is a medical image modality that uses low-coherence interferometry to visualize micro structures in biological tissue. Its current primary clinical uses are for retina and cornea health monitoring and disease diagnosis (Petzold et al., 2010; Saidha et al., 2015; Rothman et al., 2019), but different hardware configurations and scanning settings cause image variations in the forms of contrast, speckle, and noise (Wei et al., 2023). For example, Zeiss Cirrus OCT quickly scans the retina in exchange for low contrast and signal-to-noise ratio (SNR), whereas Heidelberg Spectralis OCT slowly scans the retina (via multiple image acquisitions) in exchange for superior contrast and SNR (via averaging of the multiple acquisitions). This type of *domain shift* caused by different scanners or acquisition parameters creates a challenge in the development and deployment of robust and generalizable medical image analysis methods (Guan and Liu, 2021), affecting multi-center studies and longitudinal analyses where imaging data are pooled

---

[*]Corresponding author: swei14@jhu.edu    code: https://github.com/Shuwen-Wei/lmsb

from different sources. To address domain shift, medical image harmonization, which seeks to map images from one domain to another (i.e., from Cirrus to Spectralis) while preserving the underlying anatomy, has emerged as an important preprocessing strategy (Mirzaalian et al., 2016; Zhu et al., 2017; Liu et al., 2017; Huang et al., 2018; Wrobel et al., 2020; Park et al., 2020; Zuo et al., 2021a,b; Tian et al., 2022; Cackowski et al., 2023; Zuo et al., 2023, 2024).

**Harmonization as a Coupling Problem**    Harmonization between two datasets can be formulated as a coupling problem. Let $(\Omega_i, \mathcal{F}_i, P_i)$ with $i \in \{0, 1\}$ be two probability spaces, where the $\Omega_i$'s are two sets, the $\mathcal{F}_i \subseteq 2^{\Omega_i}$'s are two $\sigma$-algebras on their respective sets, and the $P_i$'s, map the $\mathcal{F}_i$'s to $[0, 1]$, are probability measures. A coupling between these two probability spaces is the new probability space $(\Omega_{0,1}, \mathcal{F}_{0,1}, P_{0,1})$, where $\Omega_{0,1} = \Omega_0 \times \Omega_1$, and $P_{0,1} : \mathcal{F}_{0,1} \to [0, 1]$ such that $\mathrm{proj}_{0\#}P_{0,1} = P_0$ and $\mathrm{proj}_{1\#}P_{0,1} = P_1$. Here, $\mathrm{proj}_0 : \Omega_{0,1} \to \Omega_0$ and $\mathrm{proj}_1 : \Omega_{0,1} \to \Omega_1$ are defined as the projection maps, and $\#$ indicates the pushforward of the probability measure. A naive coupling is the independent coupling $P_{0,1} = P_0 \otimes P_1$, which means that given an image from one probability space, we arbitrarily sample an image from the other probability space as the harmonization result. This is obviously not a good coupling for preserving anatomy, which is undesirable because preserving anatomical information is critical for consistent analyses. To find a good coupling, additional constraints or assumptions must be applied. Generative adversarial networks (GANs) (Goodfellow et al., 2020) find a deterministic coupling by adversarial training without further constraints, and thus do not preserve anatomy. Both the optimal transport (OT) (Monge, 1781; Kantorovich, 1942; Villani, 2008) and Schrödinger bridge (SB) (Schrödinger, 1931, 1932; Léonard, 2014) approaches find an optimal coupling assuming a transport cost and a reference path measure, respectively. However, some commonly assumed transport cost and reference path measure for OTs and SBs are not correct for medical image harmonization because images with the same anatomy but different contrast do not necessarily have small Euclidean distance.

**Main Contributions**    In this paper, we propose a new method for OCT image harmonization that explicitly reduces anatomical shifts. This method, named anatomy-guided latent metric Schrödinger bridge (LMSB), follows from two key contributions. First, we provide a full analysis of the anatomy shift issue of applying SBs directly to harmonization and show several theoretical solutions to address this issue. Second, we implement our theory by training an invertible neural network (INN) that maps OCT images to a latent Euclidean metric space via anatomy guidance. We call the pullback metric as the latent metric which minimizes the distances among OCT images with the same anatomy. We then train a SB to map between distributions in this learned latent Euclidean metric space. The whole training process does not require any paired training data. We demonstrate LMSB on the harmonization of OCT images and show that it achieves better performance in anatomy preservation than conventional SBs and other baseline methods.

## 2    Related Work

**Medical Image Harmonization**    Traditional medical image harmonization methods can be divided into supervised and unsupervised methods. Although supervised harmonization methods yield good results (Tian et al., 2022; Zuo et al., 2023), the paired images they require for training are not widely available. Unsupervised harmonization methods such as CycleGAN (Zhu et al., 2017), UNIT (Liu et al., 2017), MUNIT (Huang et al., 2018), and CUT (Park et al., 2020) do not require paired data, but they generally suffer from anatomical changes during the harmonization process.

**Optimal Transport**    OT (Monge, 1781; Kantorovich, 1942; Villani, 2008) is a method to find an optimal coupling that minimizes an overall transport cost. For discrete measures, OT can be solved exactly, but it is computationally expensive. A landmark paper by (Cuturi, 2013) shows that an entropy-regularized version of OT can be solved more efficiently using the Sinkhorn algorithm (Sinkhorn, 1967). Deep learning approaches such as Wasserstein GANs (Arjovsky et al., 2017) have been established for solving OT in its Kantorovich duality form (Kantorovich, 1940). However, they are difficult to train and some constraints are difficult to implement. For example, when the transport cost is the Euclidean distance, a bounded Lipshitz continuity condition needs to be satisfied in the Kantorovich duality form (Kantorovich, 1940). But this condition can only be approximated with either spectral normalization (Miyato et al., 2018), weight clipping (Arjovsky et al., 2017), or a penalty on the gradient (Gulrajani et al., 2017). This also makes it hard to generalize to different transport costs using a deep learning framework.

**Diffusion Schrödinger Bridge** SB can be solved by iterative projection fitting (IPF) (Kullback, 1968; Rüschendorf and Thomsen, 1993; Rüschendorf, 1995), which projects to the path measure that satisfies the initial distribution and the final distribution iteratively. However, IPF is computationally expensive because of the high-dimensional nature of the path measure. Diffusion SB (DSB) (De Bortoli et al., 2021) uses a novel implementation of IPF which takes advantage of diffusion models (Sohl-Dickstein et al., 2015; Ho et al., 2020; Song et al., 2021b) by breaking down the path measure and optimizing over smaller pieces of a Markov chain through time reversal of the stochastic process (Anderson, 1982). A dual form of IPF called iterative Markovian fitting (IMF) was applied to DSB (Shi et al., 2023; Peluchetti, 2023) and has demonstrated improved performance. Compared with IPF, IMF projects the path measure into Markov measure and reciprocal class iteratively, while preserving the initial and final distribution. Due to its state-of-the-art performance, DSB (De Bortoli et al., 2021; Shi et al., 2023) has found many applications in natural images, such as image restoration by image-to-image SB ($I^2$SB) (Liu et al., 2023), and style transfer by dual diffusion implicit bridge (DDIB) (Su et al., 2023). However, DSB still introduces anatomical shifts when directly applied to medical images.

## 3 Latent Metric Schrödinger Bridge

We introduce the dynamic SB used in DSB, and show that it is equivalent to the solution of a static SB and the solution of an entropy-regularized OT. We make the observation that the assumption behind this entropy-regularized OT is not correct for medical image harmonization to preserve anatomy. We then propose the LMSB framework for harmonization, which improves anatomical consistency.

**Dynamic Schrödinger Bridge** We assume that $\Omega_0 = \Omega_1 = \mathbb{R}^d$ in the coupling problem setup in Sec. 1. Let $\Omega = C([0,1], \mathbb{R}^d)$ be all continuous $\mathbb{R}^d$-valued paths on the unit time interval $[0,1]$. We construct a probability space $(\Omega, \mathcal{F}, P)$ by adding the $\sigma$-algebra $\mathcal{F} \subseteq 2^\Omega$ and the path measure $P : \mathcal{F} \to [0,1]$. $\mathcal{P}(\Omega)$ is the set of all path measures, $\Pi(P_0, P_1) \subset \mathcal{P}(\Omega)$ the subset of path measures with their marginal densities at $t = 0$ and $t = 1$ being $P_0$ and $P_1$, respectively, and $\mathcal{M}(\Omega) \subset \mathcal{P}(\Omega)$ the subset of path measures that are Markovian. The dynamic SB problem (Schrödinger, 1931, 1932; Léonard, 2014) finds the optimal path measure $P^{\text{SB}} \in \Pi(P_0, P_1)$ with respect to a reference path measure $Q \in \mathcal{M}(\Omega)$ by minimizing their Kullback–Leibler (KL) divergence,

$$P^{\text{SB}} = \underset{P \in \Pi(P_0, P_1)}{\arg\min} \ D_{\text{KL}}(P \parallel Q), \tag{1}$$

where $Q \in \mathcal{M}(\Omega)$ is a Markov path measure of a random process $X_t$ described by the forward stochastic differential equation (SDE), $dX_t = f_t(X_t)dt + g_t dW_t$, where $X_0 \sim P_0$ without loss of generality, $f : [0,1] \times \mathbb{R}^d \to \mathbb{R}^d$ and $g : [0,1] \to \mathbb{R}$ are the drift and diffusion coefficients, and $W_t$ is the standard Wiener process.

**Static Schrödinger Bridge** A connection between the dynamic and static SB problems was established by (Föllmer, 1988). The static SB problem finds the optimal joint law $P_{0,1}^{SB}$ at times $t = 0$ and $t = 1$ by,

$$P_{0,1}^{\text{SB}} = \underset{P_{0,1} \in \Gamma(P_0, P_1)}{\arg\min} \ D_{\text{KL}}(P_{0,1} \parallel Q_{0,1}), \tag{2}$$

where $\Gamma(P_0, P_1)$ is the set of joint laws at times $t = 0$ and $t = 1$ with their marginals being $P_0$ and $P_1$, respectively, and $Q_{0,1}$ is the joint law of $Q$ at times $t = 0$ and $t = 1$. It can be shown that the solution $P^{SB}$ to Eq. 1 can be decomposed into a mixture of bridges as $P^{\text{SB}} = P_{0,1}^{\text{SB}}Q_{|0,1}$, where $P_{0,1}^{SB}$ is the solution to Eq. 2; conversely if $P_{0,1}^{\text{SB}}$ is the solution to Eq. 2, the mixture of bridges $P^{\text{SB}} = P_{0,1}^{\text{SB}}Q_{|0,1}$ is the solution to Eq. 1. Here, the mixture of bridges $P^{\text{SB}} = P_{0,1}^{\text{SB}}Q_{|0,1}$ is a short notation for $P^{\text{SB}}(\cdot) = \int_{\mathbb{R}^d \times \mathbb{R}^d} Q_{|0,1}(\cdot|x_0, x_1)P_{0,1}^{\text{SB}}(dx_0, dx_1)$, where $Q_{|0,1}$ is the diffusion bridge of $Q$ conditioned on the initial condition $x_0$ at $t = 0$ and the final condition $x_1$ at $t = 1$. Therefore, the solutions to Eq. 1 and Eq. 2 are equivalent.

**Entropy-Regularized Optimal Transport** Reference path measure $Q$ is associated with a random process $X_t$ without drift, i.e., $dX_t = g_t dW_t$ where $f_t = 0$, then Eq. 2 can be derived as,

$$P_{0,1}^{\text{SB}} = \underset{P_{0,1} \in \Gamma(P_0, P_1)}{\arg\min} \int_{\mathbb{R}^d \times \mathbb{R}^d} \|x_1 - x_0\|^2 P_{0,1}(dx_0, dx_1) - 2\sigma^2 H(P_{0,1}), \tag{3}$$

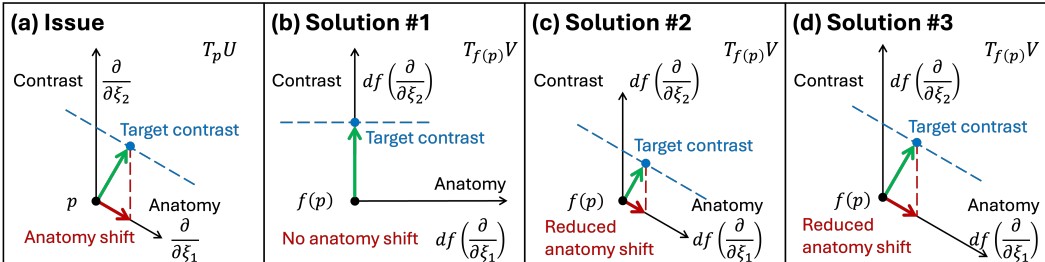

Figure 1: Local view of the anatomy and contrast geometry. The smooth map $f : U \to V$ induces a local linear map $df$ between the tangent space $T_pU$ and the tangent space $T_{f(p)}V$. The tangent vectors $\frac{\partial}{\partial \xi_1}$ and $df(\frac{\partial}{\partial \xi_1})$ point to the direction where the anatomy changes. The tangent vectors $\frac{\partial}{\partial \xi_2}$ and $df(\frac{\partial}{\partial \xi_2})$ point to the direction where the contrast changes. The blue dashed lines show the same target contrast, and they are parallel to the anatomy change direction. The green arrows connect the black points to the blue points with the shortest distance. The red dash lines are parallel to the contrast change direction, and the red arrows show the anatomy shift amount. **(a)** Anatomy shift is introduced if we use the Euclidean metric in the image intensity coordinate system. **(b)** Solution #1: find a $f$ that flattens the metric, which eliminates the anatomy shift. However, this is not always possible for an open neighborhood of $p$ for an arbitrary metric. **(c)** Solution #2: find a $f$ that shrinks the distance along the contrast direction, which reduces the anatomy shift. **(d)** Solution #3: find a $f$ that stretches the distance along the anatomy direction, which also effectively reduces anatomy shift.

where $\sigma^2 = \int_0^1 g_t^2 dt$, and $H$ is the entropy of the joint law $P_{0,1}$. We observe that the first term in Eq. 3 is the same objective as the OT formulation. Therefore, a SB with a non-drift Brownian motion as the reference path measure, is equivalent to an entropy-regularized OT problem, with the transport cost being the square of Euclidean distances. However, using this transport cost is not suitable for medical image harmonization where preserving anatomy is essential.

**Issues of Euclidean Metric and Potential Solutions**    We illustrate the issue of using a Euclidean metric in the image intensity coordinate system between two images in $\mathbb{R}^2$ as an example. Suppose $\forall p \in \mathbb{R}^2$, there is an open neighborhood $U \subseteq \mathbb{R}^2$, such that $p \in U$ and there is a coordinate system $\xi : U \to \mathbb{R}^2$ where $\xi_1$ and $\xi_2$ are the anatomy coordinate and contrast coordinate, respectively. At the same time, there is another coordinate system on $U$ which uses the image intensity value from each pixel (in this case two pixels) as a coordinate, which we denote as $\xi' : U \to \mathbb{R}^2$. It is unlikely that $\xi$ and $\xi'$ are the same. Now if we equip $U$ with the metric tensor $g' = \delta_{i'j'}d\xi^{i'} \otimes d\xi^{j'}$, i.e., an Euclidean metric on the coordinate system $\xi'$, then the tangent vectors $\frac{\partial}{\partial \xi'_1}$ and $\frac{\partial}{\partial \xi'_2}$ are orthogonal, but $\frac{\partial}{\partial \xi_1}$ and $\frac{\partial}{\partial \xi_2}$ might not be. This is illustrated in Fig. 1(a), where $\frac{\partial}{\partial \xi_1}$ and $\frac{\partial}{\partial \xi_2}$ are not orthogonal under this metric. Because $\xi_1$ and $\xi_2$ are the anatomy coordinate and contrast coordinate, $\frac{\partial}{\partial \xi_1}$ points to the direction where anatomy changes, and $\frac{\partial}{\partial \xi_2}$ points to the direction where contrast changes. If we try to find the closest point in the target contrast space (blue dashed line), we can do an orthogonal projection as shown by the green arrow, and the blue point is the solution with the shortest distance. However, it introduces an anatomy shift as indicated by the red dashed line and the red arrow.

To eliminate the anatomy shift, we need $\frac{\partial}{\partial \xi_1}$ and $\frac{\partial}{\partial \xi_2}$ to be orthogonal. It is sufficient to achieve this by choosing $g = \delta_{ij}d\xi^i \otimes d\xi^j$ as the metric tensor, and we can find a smooth map $f$ whose pushforward metric of $g$ is Euclidean metric on the new coordinate system, as shown in Fig. 1(b). This is ideal because the anatomy shift is totally eliminated. However, such map may not exist to flatten the metric on an open neighborhood of $p$, unless the Riemann curvature tensor of this metric space vanishes (Lee, 2019), which depends on how the metric tensor $g$ is defined in the open neighborhood of $p$. Alternatively, there are ways to reduce the anatomy shift. One way is to find a map $f$ that maps images into a latent Euclidean metric space, whose pullback metric shrinks the distances along the contrast direction. As shown in Fig. 1(c), the target contrast moves closer to the black point, which reduces the anatomy shift. Another way is to find a map $f$ that maps images into a latent Euclidean metric space, whose pullback metric stretches the distances along the anatomy direction. As shown

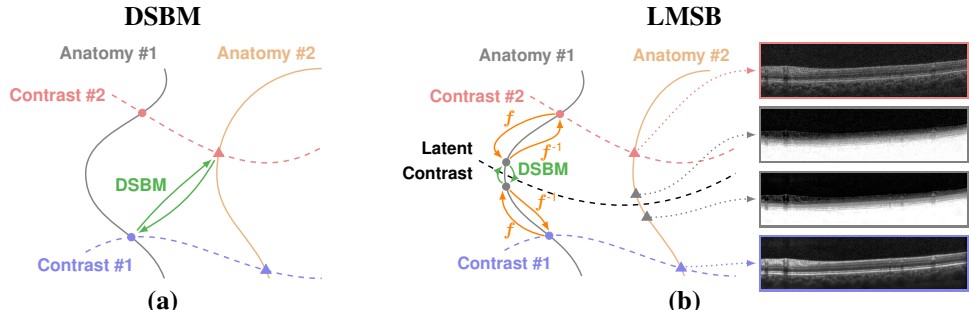

Figure 2: Global view of the anatomy and contrast geometry. **(a)** Diffusion Schrödinger Bridge Matching (DSBM): Solid lines indicate the same anatomy and dashed lines indicate the same contrast. Directly applying DSBM between Contrast #1 and #2 results in a shift in anatomy, as shown by the green arrows. **(b)** Latent Metric Schrödinger Bridge (LMSB): A Latent Contrast space is constructed. An invertible $f$ is trained to map both contrasts to the Latent Contrast, as shown by the orange arrows. Applying DSBM to anatomies within the Latent Contrast reduces anatomical shift. The right panel of OCT images show examples of the same anatomy in Contrast #1 (bottom) and #2 (top), as well as the anatomy within the Latent Contrast.

in Fig. 1(d), in this case, although the target contrast space does not get closer, the anatomy shift is still effectively reduced, by comparing the red arrow with the tangent vector $df(\frac{\partial}{\partial \xi_1})$.

In practice, medical images live in a higher dimension space. It is almost impossible to decouple the coordinates into an anatomy component and a contrast component if we use the intensity values in each pixel as the coordinate system for medical images. Therefore, if we directly apply SBs for medical image harmonization, or equivalently entropy-regularized OTs with the square of Euclidean distances as the transport cost, it is very likely that they shift the anatomy. Therefore, it is not a good idea to use the square of Euclidean distances as the transport cost in the OT formulation. A potential solution is to use an alternative transport cost, such as local normalized cross correlation (LNCC), which is often used in medical image registration to evaluate the alignment of anatomical structures (Chen et al., 2025). The LNCC compresses the distances along the contrast direction, as shown by $df(\frac{\partial}{\partial x_2})$ in Fig. 1(c), and thus effectively reduces anatomical shift even if it exists. However, an OT problem with an arbitrary transport cost is difficult to solve using deep learning frameworks, and thus is not practically useful. Another alternative method is to find a coordinate transformation of a metric space whose pushforward metric is the Euclidean metric, and then do the OT in the new coordinate system. However, this is generally not possible because an arbitrary metric space is generally not equivalent to the Euclidean metric globally or even locally in an open neighborhood, unless the Riemann curvature tensor of this metric space vanishes (Lee, 2019). However, it still inspires our proposed method LMSB, which tries to find a coordinate transformation to compress the distance along the contrast direction, which we explain in the following section.

**Latent Metric Schrödinger Bridge**  The core idea of the LMSB is to find an invertible map $f$ that maps the images into a latent Euclidean metric space, where the distances among the images with the same anatomy are minimized using the pullback latent metric. Then, we use the DSB matching (DSBM) to match the distribution in the new coordinate system to minimize the anatomical shift. The difference between DSBM and LMSB is shown in Fig. 2. The solid lines indicate the same anatomy. The dashed lines indicate the same contrast.

**Invertible Neural Network**  We build a deep learning based invertible network $f$ using the idea of affine coupling that is implemented in normalizing flow (Dinh et al., 2015; Rezende and Mohamed, 2015; Dinh et al., 2017; Kingma and Dhariwal, 2018; Stimper et al., 2023). The structure of the invertible network is shown in Fig. 3. The network first splits the input images $w$ into two parts $w = [w_c, w_{\bar{c}}]$ through a checkerboard decomposition. Then the affine coupling layers map them to $[z_c, z_{\bar{c}}] = [w_c + f_2(w_{\bar{c}} + f_1(w_c)), w_{\bar{c}} + f_1(w_c)]$. In the end, $[z_c, z_{\bar{c}}]$ are merged to get the final output $z = [z_c, z_{\bar{c}}]$. The affine coupling layers $f_1$ and $f_2$ use two U-Nets (Ronneberger et al., 2015) with the same structure but different weights. Both the split and merge operations are invertible. For the affine coupling layers, it can be inverted by $[w_c, w_{\bar{c}}] = [z_c - f_2(z_{\bar{c}}), z_{\bar{c}} - f_1(z_c - f_2(z_{\bar{c}}))]$.

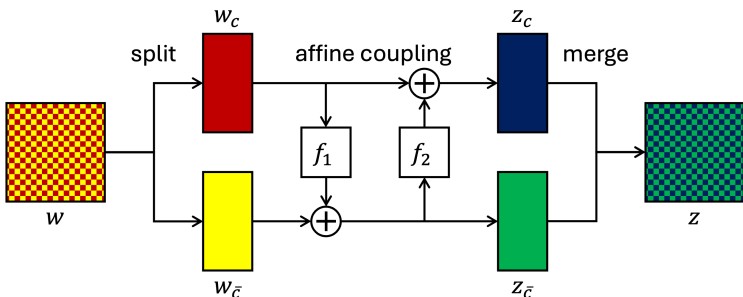

Figure 3: Structure of the invertible neural network $f$. The input $w$ is split into two parts $[w_c, w_{\bar{c}}]$ through a checkerboard decomposition, and goes through two affine coupling layers $f_1$ and $f_2$ to generate $[z_c, z_{\bar{c}}]$, and merges together to produce the output $z$.

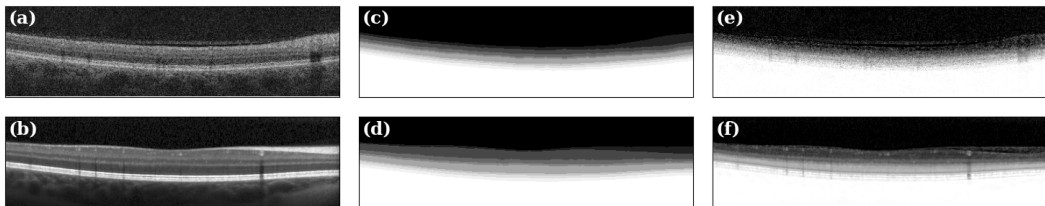

Figure 4: **(a)** Cirrus OCT B-scan. **(b)** Spectralis OCT B-scan. **(c)** Segmentation label map of **(a)**. **(d)** Segmentation label map of **(b)**. **(a, c)** and **(b, d)** are used as constructed paired images $(w, z)$ for training Eq. 6. After training, the invertible map takes **(a)** and **(b)** to **(e)** and **(f)**, respectively.

**Training Objective**    To train the invertible map $f$, we first assume that we have paired images $\{(u_i, v_i)\}_{i=1}^n$ with the same anatomy but different contrasts, so that we can minimize the following training objective,

$$\arg\min_{f} \mathbb{E}_{u,v}\Big[\|f(u) - f(v)\|\Big], \tag{4}$$

where $(u, v)$'s are randomly sampled from the paired images $\{(u_i, v_i)\}_{i=1}^n$. However, in practice, we only have unpaired images $\{u_i\}_{i=1}^{n_1}$ and $\{v_i\}_{i=1}^{n_2}$, and we do not want a less general method that requires paired images. To solve the issue, we note the training objective in Eq. 4 is bounded by,

$$\mathbb{E}_{u,v}\Big[\|f(u) - f(v)\|\Big] \leq \mathbb{E}_{u,z}\Big[\|f(u) - z\|\Big] + \mathbb{E}_{v,z}\Big[\|f(v) - z\|\Big], \tag{5}$$

where we add a latent variable $z$ of the same dimension to each image pair $(u, v)$, we use the triangle inequality $\|f(u) - f(v)\| \leq \|f(u) - z\| + \|f(v) - z\|$, and we marginalize the unrelated random variables. Therefore, if we find a common $z$ for images with the same anatomy but different contrasts, we can train the invertible map $f$ by an alternative training objective,

$$\arg\min_{f} \mathbb{E}_{w,z}\Big[\|f(w) - z\|\Big], \tag{6}$$

where $(w, z)$'s are randomly sampled from a constructed paired dataset $\{(w_i, z_i)\}_{i=1}^{n_1+n_2}$, in which $w$'s are from $\{u_i\}_{i=1}^{n_1} \cup \{v_i\}_{i=1}^{n_2}$, and $z$ is constructed for each $w$. There are many choices for $z$ when building the paired dataset $\{(w_i, z_i)\}_{i=1}^{n_1+n_2}$. A sufficient condition for Eq. 5 to hold is that $z$ has the same anatomy as $w$ within a latent contrast, so that images with the same anatomy but different contrasts all map close to the same $z$. We call the space of $z$ the Latent Contrast space, as shown in Fig. 2(b). A straightforward design is to use a segmentation label map of $w$ as $z$, where we assign an unique intensity to different pixel labels. Examples of paired $w$'s and $z$'s are shown in Fig. 4. Intuitively, the invertible map $f$ brings both Cirrus and Spectralis B-scans closer to a common latent contrast while preserving the anatomy using segmentation label maps so that the anatomy shift, i.e., segmentation change, is reduced during harmonization.

# 4 Experiments

**Dataset**    The OCT dataset consists of 388 Cirrus volumes and 338 Spectralis volumes. The 388 Cirrus volumes come from 194 subjects (388 eyes in total). The 338 Spectralis volumes come from 165 subjects (269 eyes in total). Note that there are repeated scans on the same eye for Spectralis. Training and testing splits are done by subject: 352 Cirrus volumes from 176 subjects (352 eyes) and 307 Spectralis volumes from 156 subjects (252 eyes) for training and 36 Cirrus volumes from 18 subjects (36 eyes) and 31 Spectralis volumes from 9 subjects (17 eyes) for testing. Our experiments are in 2D and operate on B-scans independently. Because each Cirrus volume contains 128 B-scans and each Spectralis OCT volume contains 49 B-scans over the same field of view $6 \times 6 \ mm^2$, the Cirrus volumes have denser B-scan sampling. Therefore, to reduce anatomical redundancy across individual Cirrus volume B-scans, we extract every third B-scan. Specifically, in the training dataset, we extract $15,000$ B-scans in the training dataset and $1,500$ B-scans in the testing dataset for both Cirrus and Spectralis, resulting in $30,000$ training B-scans and $3,000$ testing B-scans with no subject data leakage between train and test splits.

The original size of Cirrus B-scans is $1024 \times 512$ (axial $\times$ lateral) and the original size of Spectralis B-scans is $496 \times 1024$ (axial $\times$ lateral). The axial resolution of Cirrus B-scans is $2 \ \mu m/pixel$, and the axial resolution of Spectralis B-scans is $4 \ \mu m/pixel$. Both B-scans cover a $6 \ mm$ lateral scanning range. We crop the Cirrus B-scans axially to $512 \times 512$ and resize them to $128 \times 512$, and we crop the Spectralis B-scans axially to $256 \times 1024$ and resize them to $128 \times 512$, such that they have the same image size, digital resolution, and field of view.

**Network Training**    We trained the invertible neural network $f$ with $15,000$ Cirrus and $15,000$ Spectralis OCT B-scans in the training set. We constructed a paired dataset $\{(w_i, z_i)\}_{i=1}^{30,000}$ by generating a segmentation label map $z$ for each B-scan $w$ using a deep learning based retinal OCT segmentation algorithm (He et al., 2019, 2021, 2023). Examples of these paired images are shown in Fig. 4, where a Cirrus OCT B-scan and its segmentation label map is shown in Fig. 4(a) and Fig. 4(c), respectively, and a Spectralis OCT B-scan and its segmentation label map is shown in Fig. 4(b) and Fig. 4(d), respectively. We then trained $f$ using this constructed paired dataset by optimizing the training objective in Eq. 6. There are two training strategies we used to make the invertible network more robust. First, we augmented the paired dataset by adding Gaussian noise to $w$, with the paired $z$ being unchanged. Second, we constrained the space of the invertible map to volume preserving maps, i.e., Jacobian determinant of $f$ is 1 everywhere. We note that this constraint was imposed by our network structure in Fig. 3. Training results are shown in Fig. 4(e) and Fig. 4(f) for the Cirrus and Spectralis OCT B-scans, respectively. Both the Cirrus and Spectralis OCT B-scans are not perfectly mapped to the segmentation label maps, and some detailed anatomical structures such as vessel and shadows are preserved. This is a desirable property because perfectly mapping into the segmentation label map will result in poor invertibiliy. Without the use of the noise augmentation and volume preserving map, we will not get this property because the network will overfit on the training data.

We trained both a DSB in the original image domain, which we denote as DSBM, and in the latent contrast space which we denote as LMSB. We adapted the implementation in DSBM (Shi et al., 2023) for both methods and use the exact same parameters for a fair comparison. Specifically, we ran 30 steps of IMF in total to solve DSBM and LMSB. For the first IMF step, we used independent coupling as the intial coupling, and we ran $10,000$ iterations both forward and backward. For the remaining IMF steps, we ran $2,500$ iterations both forward and backward, and we cached simulation trajectories every $1,250$ iterations. We set the number of diffusion steps to 100. We let the diffusion coefficient be constant at each diffusion step and choose $\sigma^2 = 0.1$. After training, we compared their performance with two sampling strategies: 1) stochastic differential equation (SDE); and 2) ordinary differential equation (ODE). The difference between SDE and ODE is that SDE preserves the path measure of a stochastic process, but ODE only preserves the marginal density of the path measure. Furthermore, ODE is deterministic and invertible, which is desirable in many applications, but SDE is stochastic and not invertible.

We also trained a dual diffusion implicit bridge (DDIB) (Su et al., 2023) as another method for comparison. To map between two distributions using a DDIB, we trained two independent denoising diffusion probabilistic models (DDPMs) (Ho et al., 2020) on both Cirrus and Spectralis training sets with $1,000$ diffusion steps. After training, we use the denoising diffusion implicit model (DDIM) (Song et al., 2021a) sampling strategy to convert an image from one domain to a latent Gaussian noise, and then to an image in another domain. DDIM can also be categorized as an

Table 1: Mean MAE (Std. Dev.) comparison ($N = 1,500$) across nine retinal boundaries for both Spectralis to Cirrus and Cirrus to Spectralis harmonization. Bold numbers indicate the best result in that row for that subtask. Asterisks indicate statistical significance (i.e., paired t-test comparing 1st and 2nd best results from the other two methods gave p-value $< 0.05$). **Key:** ILM: internal limiting membrane; RNFL: retinal nerve fiber layer; GCL: ganglion cell layer; IPL: inner plexiform layer; INL: inner nuclear layer; OPL: outer plexiform layer; ONL: outer nuclear layer; ELM: external limiting membrane; IS: inner segment; OS: outer segment; RPE: retinal pigment epithelium complex; BM: Bruch's membrane; AVG: Average.

| | Spectralis to Cirrus | | | | | Cirrus to Spectralis | | | | |
| --- | --- | --- | --- | --- | --- | --- | --- | --- | --- | --- |
| | **DDIB** | **DSBM** | | **LMSB** | | **DDIB** | **DSBM** | | **LMSB** | |
| | ODE | SDE | ODE | SDE | ODE | ODE | SDE | ODE | SDE | ODE |
| ILM | 0.82 (0.79) | 0.47 (0.23) | 0.50 (0.36) | 0.27 (0.28) | **0.26**\* (0.32) | 1.29 (2.35) | 0.64 (1.01) | 0.61 (0.85) | 0.29 (0.48) | **0.23**\* (0.52) |
| RNFL -GCL | 1.19 (0.95) | **0.73**\* (0.29) | 0.75 (0.34) | 0.89 (0.46) | 0.88 (0.45) | 1.64 (2.41) | 0.71 (1.98) | **0.69** (2.33) | 0.78 (0.51) | 0.73 (0.54) |
| IPL -INL | 0.96 (0.74) | 0.75 (0.32) | 0.71 (0.33) | 0.62 (0.33) | **0.54**\* (0.34) | 1.22 (1.91) | 0.66 (1.91) | 0.61 (2.15) | 0.55 (0.59) | **0.47**\* (0.56) |
| INL -OPL | 0.90 (0.72) | 0.52 (0.22) | 0.47 (0.23) | 0.45 (0.24) | **0.39**\* (0.27) | 1.06 (1.56) | 0.52 (1.59) | 0.46 (1.85) | 0.43 (0.74) | **0.34**\* (0.68) |
| OPL -ONL | 0.96 (0.72) | 0.66 (0.26) | 0.64 (0.26) | 0.52 (0.26) | **0.48**\* (0.29) | 1.09 (1.29) | 0.60 (1.29) | 0.54 (1.44) | 0.50 (0.89) | **0.41**\* (0.84) |
| ELM | 0.71 (0.74) | **0.29**\* (0.16) | 0.29 (0.19) | 0.30 (0.17) | 0.31 (0.22) | 0.83 (0.99) | 0.35 (0.81) | 0.33 (0.84) | 0.33 (1.12) | **0.30**\* (1.11) |
| IS -OS | 0.70 (0.75) | 0.26 (0.14) | **0.25** (0.17) | 0.27 (0.15) | 0.26 (0.22) | 0.81 (0.94) | 0.32 (0.67) | 0.30 (0.68) | 0.30 (1.20) | **0.27** (1.20) |
| OS -RPE | 0.81 (0.74) | 0.34 (0.12) | 0.35 (0.15) | **0.31**\* (0.16) | 0.31 (0.27) | 0.85 (0.94) | 0.34 (0.65) | 0.33 (0.62) | 0.30 (1.25) | **0.28**\* (1.20) |
| BM | 0.93 (0.75) | 0.45 (0.15) | 0.44 (0.18) | 0.43 (0.23) | **0.41**\* (0.33) | 0.93 (0.93) | 0.40 (0.61) | **0.38** (0.58) | 0.42 (1.17) | 0.39 (1.12) |
| **AVG** | 0.89 (0.65) | 0.50 (0.13) | 0.49 (0.17) | 0.45 (0.18) | **0.43**\* (0.21) | 1.08 (1.37) | 0.51 (1.16) | 0.47 (1.24) | 0.43 (0.86) | **0.38**\* (0.84) |

ODE sampling strategy. In practice, we do not go through all $1,000$ diffusion steps, but stop early at 500 steps for a balance between sampling quality and harmonization quality (Wei et al., 2026).

**Anatomy Consistency Comparison** We compared the performance of our proposed LMSB with the DSBM and the DDIB by evaluating the anatomical consistency before and after harmonization. To do so, we ran all the methods on the testing dataset that contains $1,500$ Cirrus B-scans and $1,500$ Spectralis OCT B-scans to generate Cirrus B-scans from Spectralis B-scans and vice versa as two harmonization tasks. We then applied the deep learning based retinal OCT segmentation method (He et al., 2019, 2021, 2023) to identify nine retinal boundaries. We computed the mean absolute error (MAE) for these boundary locations before and after harmonization. The results are summarized in Table 1. We see that LMSB ODE achieves the best performance in preserving anatomy during harmonization.

MAE evaluation may be subject to segmentation bias or errors. Therefore, we computed the LNCC between the OCT images before and after harmonization to evaluate structural similarity, which is shown in Table 2. Higher LNCC means higher structural similarity. We see that for both harmonization subtasks, LMSB outperforms DSBM with either SDE or ODE as sampling strategies, and outperforms DDIB with ODE sampling strategy. Moreover, when comparing between SDE and ODE sampling strategies, we find that ODE sampling preserves the anatomy much better than SDE. We believe this is due to the speckle in OCT images, which is preserved better by ODE.

Table 2: Mean LNCC (Std. Dev.) comparison ($N = 1,500$) on both Spectralis to Cirrus and Cirrus to Spectralis harmonization. Bold numbers indicate the best result in that row for that subtask. Asterisks indicate statistical significance (i.e., paired t-test comparing 1st and 2nd best results gave p-value $< 0.05$).

| | Spectralis to Cirrus | | | | | Cirrus to Spectralis | | | | |
|---|---|---|---|---|---|---|---|---|---|---|
| | **DDIB** | **DSBM** | | **LMSB** | | **DDIB** | **DSBM** | | **LMSB** | |
| | ODE | SDE | ODE | SDE | ODE | ODE | SDE | ODE | SDE | ODE |
| LNCC | 0.65 (0.08) | 0.16 (0.01) | 0.63 (0.05) | 0.17 (0.02) | **0.66**\* (0.05) | 0.63 (0.05) | 0.16 (0.02) | 0.67 (0.04) | 0.16 (0.02) | **0.69**\* (0.03) |

We also show a qualitative comparison in Fig. 5. The bottom rows show enlarged images of the region indicated by the yellow rectangle in the top two rows. The red solid lines and the yellow dashed lines show the retinal boundary segmentation on the original image and the harmonized image, respectively. The yellow arrows show locations where segmentation on the harmonized anatomy disagrees with the original segmentation. We see that for LMSB, the yellow dashed lines are closer to the red solid lines, which mean LMSB preserves the anatomy better than DSBM and DDIB during harmonization.

## 5 Discussion

There are potential limitations in the LMSB approach. First, the theory shows that LMSB can reduce anatomical changes during harmonization, but may not totally eliminate them (Solution #1) as discussed in Fig. 1. This is because not all metric spaces are isomorphic to Euclidean metric spaces. A potential solution is to find a coordinate transform that flattens the metric as much as possible, which we have not explored.

Second, the method constructs a reference anatomy with a latent contrast by using the segmentation label map, but it is not clear whether this latent contrast space is optimal. Specifically, as discussed in Fig. 1, we can either compress the distance along the contrast direction (Solution #2) or stretch the distance along the anatomy direction (Solution #3) to reduce the anatomical shift. We chose to compress the distance along the contrast direction, but we have not explored how to stretch the distance along the anatomy direction. For example, the segmentation label map used in this paper assigns unique numbers to different retinal layers, and the number is increasing from the top layer to the bottom layer. This is not necessarily optimal. To further stretch the distance along the anatomy direction, we can change the segmentation label maps such that the label number differences between adjacent retinal layers become larger, which effectively stretches the distance along the anatomy direction. It remains to be investigated whether this helps further improve the performance of LMSB.

Third, a different segmentation bias for Cirrus and Spectralis retinal OCT images may affect the LMSB performance when constructing the latent contrast space because it violates the assumption in Eq. 5, i.e., the latent variable $z$ should be the same for an image pair with the same anatomy but different contrast. Note that a consistent segmentation bias for both Cirrus and Spectralis, on the other hand, does not affect the LMSB performance.

Fourth, we have not evaluated the LMSB performance using real paired B-scans because individual B-scans are not well aligned even for paired OCT volumes due to the difficulty of optical alignment during acquisition. Moreover, registering two OCT volumes is not straightforward (Chen et al., 2014; Reaungamornrat et al., 2018). This is because deformable registration is required due to different geometrical distortions between Cirrus and Spectralis, and highly anisotropic digital resolutions and field strength variations further complicate the registration process. There are work that demonstrated improved OCT volume interpolation along the slow axis using deformable registration and generative models (Wei et al., 2025), which may benefit OCT volume registration but it remains to be investigated. Further developments on registering two OCT volumes and evaluations on real paired B-scans should be conducted in the future.

Fifth, we have only demonstrated the LMSB performance using retinal OCT images, which have a relatively simple anatomical structure. Theoretically, LMSB can generalize to other imaging modalities because the training of LMSB only requires unpaired datasets and corresponding segmentation

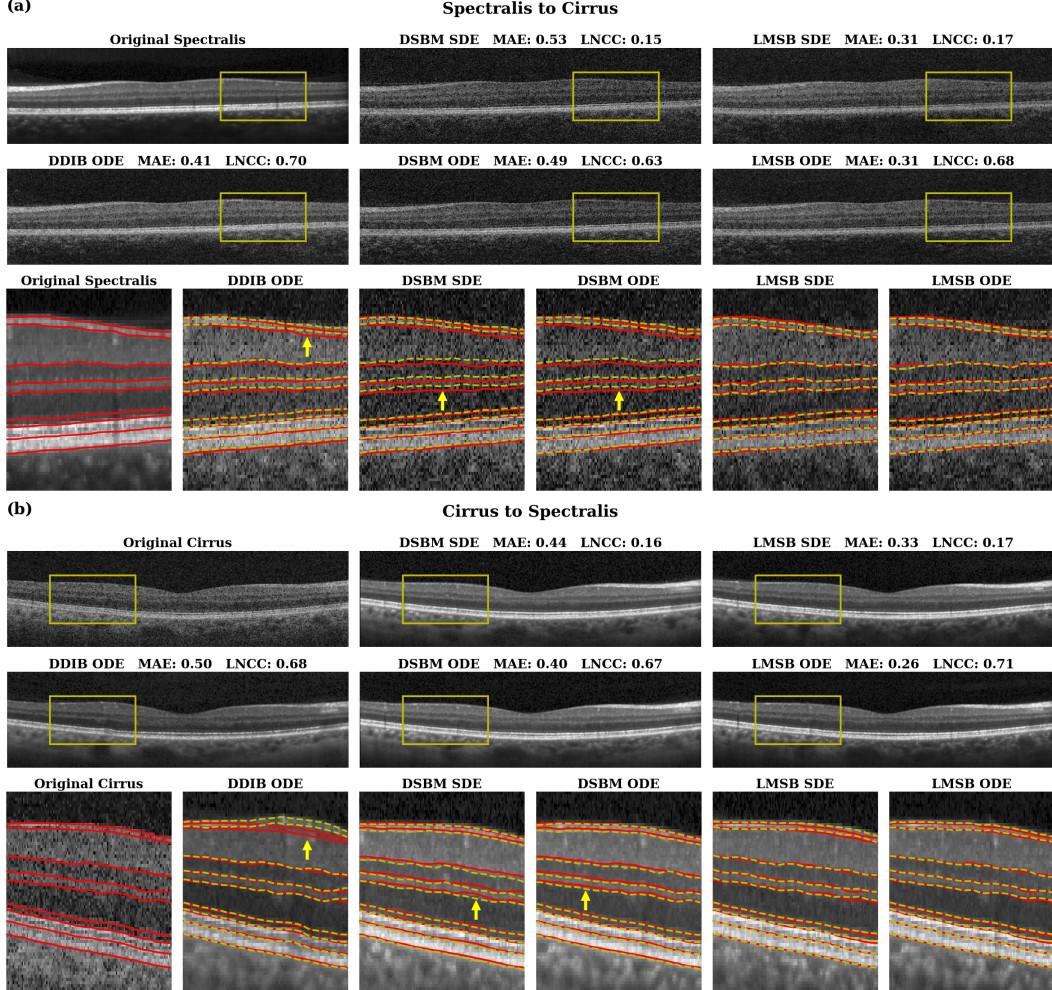

Figure 5: Qualitative comparison between DDIB, DSBM, and LMSB for **(a)** Spectralis to Cirrus harmonization and **(b)** Cirrus to Spectralis harmonization. The bottom rows show enlarged images of the region indicated by the yellow rectangle in the top two rows. The red solid lines show the retinal boundary segmentation on the original anatomy. The yellow dashed lines show the retinal boundary segmentation on the harmonized anatomy by different methods. The yellow arrows show locations where segmentation on the harmonized anatomy disagrees with the original segmentation.

label maps. However, it may require different preprocessing steps for other imaging modalities. The segmentation label map may require more careful design when the anatomy becomes more complex. Further experiments involving more complex anatomies and other imaging modalities should be studied in the future.

## 6   Conclusion

In this paper, we proposed an anatomy-guided latent metric Schrödinger bridge (LMSB) framework to improve the anatomical consistency for medical image harmonization. We trained an invertible network that maps OCT images into a latent Euclidean metric space where the distances among medical images with the same anatomy are minimized using the pullback latent metric. A diffusion Schrödinger bridge is then trained to match the distribution in this learned latent Euclidean metric space. We demonstrated our method on OCT image. We showed that the proposed LMSB method achieves a better harmonization performance while preserving the anatomy than directly applying the Schrödinger bridge and other unsupervised harmonization methods.

## Acknowledgments and Disclosure of Funding

This work is supported in part by the NIH through NEI grant R01-EY032284 (PI: J.L. Prince), NINDS grant R01-NS082347 (PI: P.A. Calabresi), as well as NIA grants R01-AG021155 (PI: S.C. Johnson) and R01-AG027161 (PI: S.C. Johnson). This material is partially supported by the National Science Foundation grant number 2136228 (PI: J. Gopalakrishnan), Graduate Research Fellowship grant number DGE-1746891 (S.W. Remedios), and a Johns Hopkins University Discovery Grant (PI: A. Carass).

J.L. Prince and A. Carass have received royalties from JuneBrain Inc. P.A. Calabresi is PI on grants to JHMI from Genentech and the Myelin Repair Foundation, and has received personal consulting Honoria from Idorsia, Spolia Therapeutics, Novartis, and Lilly. The other authors have no competing interests to declare.

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

# Appendices

# A  Proof of Volume Preservation

In this section, we provide a proof that the invertible neural network (INN) $f$ that we constructed is volume-preserving. We note that the determinant of the induced local linear map $df$ can be derived as,

$$\det(df) = \det \begin{pmatrix} \mathrm{id} & df_2 \\ 0 & \mathrm{id} \end{pmatrix} \det \begin{pmatrix} \mathrm{id} & 0 \\ df_1 & \mathrm{id} \end{pmatrix} = 1, \tag{7}$$

where $\mathrm{id}$ is the identity map, $f_1$ and $f_2$ are maps in the affine coupling layers in the INN, $df_1$ and $df_2$ are the induced local linear maps from $f_1$ and $f_2$, and we abuse the use of the notations $\mathrm{id}$, $df_1$ and $df_2$ to be the corresponding matrix representations. Because the determinant of $df$ is 1 everywhere, the invertible map $f$ is volume-preserving.

# B  Ablation Study

For the INN, we constrained the invertible map to be volume-preserving and we augmented the input images by adding Gaussian noise during network training. To demonstrate the importance of the volume preservation and the data augmentation, we conducted the following ablation studies. The first ablation implements the INN without volume preservation, but with data augmentation, which we denote as w/o Volume Preservation. This is done by adding a scaling operation to each affine coupling layer. The second ablation implements the INN without data augmentation, but with volume preservation, which we denote as w/o Data Augmentation. We denote LMSB-INN (Ours) as the implementation of the INN with both volume preservation and data augmentation.

## B.1  Comparison of INN Outputs

The INN training results of these ablation experiments are summarized in Table 3. We calculated the root mean square error (RMSE) between the outputs of the INNs and the segmentation label maps. We also calculated the local normalized cross correlation (LNCC) between the outputs of the INNs and the original input images. Lower RMSEs indicate that the INN outputs are closer to the segmentation label maps. Higher LNCCs indicate that the INN outputs have more structure similarity to the original input images. From Table 3, we see that without volume preservation, the INN produces results that are almost identical to the segmentation label maps, but have a low structural similarity to the original input images. This will lead to a poor invertibility of the INN, which we will demonstrate later. The volume preservation constraint helps improve the structural similarity between the INN outputs and the original input images. Moreover, with both volume preservation and data augmentation, the INN maps images to a neighborhood of the segmentation label map, and the INN outputs have the highest structural similarity with the original input images. Furthermore, by comparing the INN results between Spectralis and Cirrus, we see that INN produces noiser outputs with a higher structure similarity when using Cirrus B-scans as input.

We show some qualitative results in Fig. 6. Column 1 shows the input images to the INNs, including two Spectralis B-scans in Rows 1–2 and two Cirrus B-scans in Rows 3–4. Columns 2–4 show the output images of the INNs trained from w/o Volume Preservation, w/o Data Augmentation, and LMSB-INN (Ours), respectively. Column 2 shows almost identical results to the segmentation label maps that we constructed for the input images. This means that without volume preservation, the trained INN maps images with the same anatomy but different contrasts to almost the same output, i.e., the segmentation label map. The results in Column 3 shows more different output images between using Cirrus B-scans and Spectralis B-scans as input. The INN outputs are much noiser when using the Cirrus B-scans as inputs than using the Spectralis B-scans as input. This improves the invertibility of the INN. However, there are still a lot of detailed anatomical structures missing from the results in Column 3, for example, vessels and shadows. This is because the segmentation label maps that we constructed only contain the segmentation of different retinal layers, and do not include segmentation of vessels and shadows. Therefore, the constructed segmentation label maps are not perfect anatomical representations of the input images. This issue is alleviated by adding Gaussian noise to the input images as data augmentation, because it reduces the detailed anatomical structures from the original input images and makes the corresponding segmentation label maps more representative. We can see the results of LMSB-INN (Ours) with both volume preservation and data augmentation from Column 4, where the detailed anatomical structures of the vessels and shadows are well preserved for both the Cirrus and Spectralis input images.

Table 3: Mean RMSE (Std. Dev.) and mean LNCC (Std. Dev.) comparison ($N = 1,500$) on the results of the INNs trained from w/o Volume Preservation, w/o Data Augmentation, and LMSB-INN (Ours), respectively. Examples of the INN output images are shown in Fig. 6. The RMSEs were calculated between the outputs of the INNs and the target segmentation label maps. Lower RMSEs indicate that the INN outputs are closer to the segmentation label maps. The LNCCs were calculated between the outputs of the INNs and the original input images. Higher LNCCs indicate that the INN outputs have more structural similarity as the original input images.

|  |  | w/o Volume Preservation | w/o Data Augmentation | LMSB-INN (Ours) |
|---|---|---|---|---|
| **Spectralis** | RMSE | 0.013 (0.002) | 0.029 (0.005) | 0.056 (0.006) |
|  | LNCC | 0.205 (0.012) | 0.412 (0.066) | 0.540 (0.080) |
| **Cirrus** | RMSE | 0.014 (0.008) | 0.063 (0.010) | 0.078 (0.007) |
|  | LNCC | 0.102 (0.011) | 0.596 (0.011) | 0.794 (0.020) |

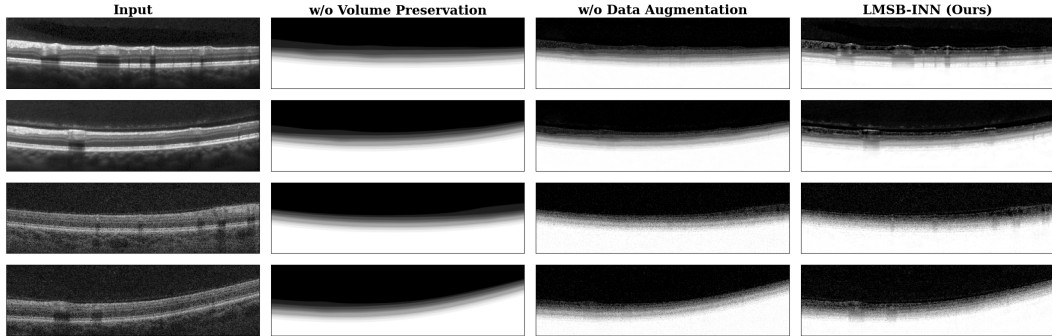

Figure 6: INN outputs. Column 1 shows the input images to the INNs, including two Spectralis B-scans in Rows 1–2 and two Cirrus B-scans in Rows 3–4. Columns 2–4 show the output images of the INNs trained from w/o Volume Preservation, w/o Data Augmentation, and LMSB-INN (Ours), respectively.

## B.2 Comparison of INN Invertibility

To test the invertibility of the INNs, we added small Gaussian noise with a standard deviation of 0.05 to the INN output images, and then applied the inverse of the INN to obtain the corresponding reconstructed images. The reconstruction results are summarized in Table 4. We calculate the RMSE and the LNCC between the reconstructed images and the original input images. Lower RMSEs and higher LNCCs indicate a better invertibility of the INNs. From Table 4, we see that without volume preservation, the RMSEs are big and the LNCCs are small, which indicates a poor invertibility. The volume preservation constraint helps improve the invertibility. Moreover, with both volume preservation and data augmentation, the reconstructed images achieve the lowest RMSE and the highest LNCC. Furthermore, the RMSEs for the LMSB-INN (Ours) are 0.05, which is identical to the standard deviation of the added Gaussian noise. This suggests that a well regularized invertibility is achieved when using both volume preservation and data augmentation.

We also show some qualitative results in Fig. 7. Column 1 shows the input images to the INN, the same as Column 1 in Fig. 6. Columns 2–4 show the reconstructed images of the INNs trained from w/o Volume Preservation, w/o Data Augmentation, and LMSB-INN (Ours), respectively. From the results in Column 2, we see that the reconstruction results are very different from the original input images in Column 1. Here, Column 2 uses the same colorbar as Column 1 for consistent comparison. This indicates that without volume preservation, the invertibility INN is diminished, where a small error in the latent space leads to a big error in the image space. The results in Columns 3–4 show a better invertibility. This is because with volume preservation, the INN establishes a bijection between a neighborhood in the image space and a neighborhood in the latent space with the same volume size, which makes the invertibility of the INN more robust. Comparing the results in Columns 3–4, we see

Table 4: Mean RMSE (Std. Dev.) and mean LNCC (Std. Dev.) comparison ($N = 1,500$) on the invertibility of the INNs trained from w/o Volume Preservation, w/o Data Augmentation, and LMSB-INN (Ours), respectively. We added small Gaussian noise with a standard deviation of $0.05$ to the INN output images, and then applied the inverse of the INNs to obtain the corresponding reconstructed images. Examples of the reconstructed images are shown in Fig. 7. Both the RMSEs and LNCCs were calculated between the reconstructed images and the original input images. Lower RMSEs and higher LNCCs indicate a better invertibility of the INNs. Bold numbers indicate the best result in that row. Asterisks indicate statistical significance (i.e., paired t-test comparing 1st and 2nd best results gave p-value $< 0.05$).

|  |  | w/o Volume Preservation | w/o Data Augmentation | LMSB-INN (Ours) |
|---|---|---|---|---|
| **Spectralis** | RMSE | 44.420 (1.138) | 0.106 (0.011) | **0.050**\* (0.000) |
|  | LNCC | 0.029 (0.002) | 0.289 (0.061) | **0.418**\* (0.057) |
| **Cirrus** | RMSE | 44.680 (1.241) | 0.070 (0.007) | **0.050**\* (0.000) |
|  | LNCC | 0.026 (0.001) | 0.496 (0.031) | **0.618**\* (0.047) |

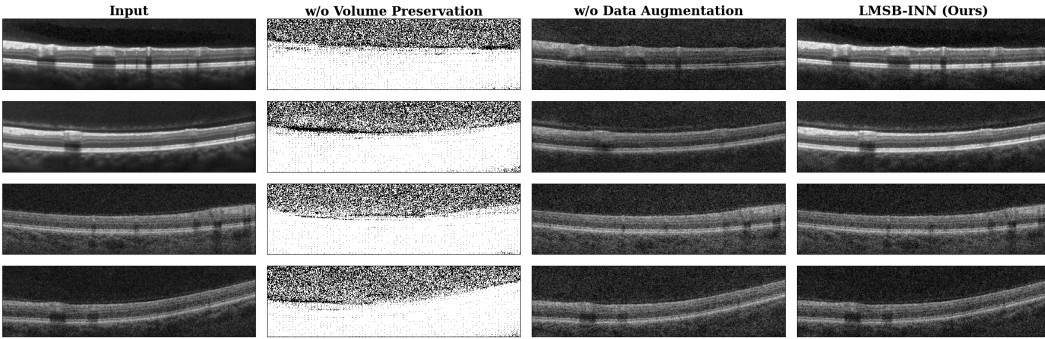

Figure 7: INN invertibility. Column 1 shows the input images to the INNs, the same as Column 1 in Fig. 6. We added small Gaussian noise with a standard deviation of $0.05$ to the INN output images that are shown in Columns 2–4 in Fig. 6, and then applied the inverse of the INNs to obtain the corresponding reconstructed images. Columns 2–4 show the reconstructed images of the INNs trained from w/o Volume Preservation, w/o Data Augmentation, and LMSB-INN (Ours), respectively.

that the invertibility is further improved with data augmentation. This is because by adding Gaussian noise to the input images as data augmentation, it provides the information where the neighborhood should be centered.

### B.3  Comparison of Anatomy Preservation

We evaluate the anatomy preservation of LMSB harmonization using the INNs trained from w/o Volume Preservation, w/o Data Augmentation, and LMSB-INN (Ours), respectively. For each INN, we trained a LMSB using the same training parameters as we described in Network Training. The harmonization results are summarized in Table 5. We calculated the LNCCs between harmonized and original images. Higher LNCCs indicate a better preservation of the anatomical structure. From Table 5, we see that without volume preservation, the harmonization results are poor. The volume preservation helps improve the anatomy preservation. Moreover, with both volume preservation and data augmentation, the harmonized images achieve the highest LNCC, for both Spectralis to Cirrus and Cirrus to Spectralis harmonization, and for both SDE and ODE sampling strategies.

We also show some qualitative results in Fig. 8. Column 1 shows the original input images before harmonization, the same as Column 1 in Fig. 6. Rows 1–2 show two examples of Spectralis to Cirrus harmonization, and Rows 3–4 show two examples of Cirrus to Spectralis harmonization. From the results in Column 2, we see that without volume preservation, the poor invertibility leads to a bad harmonization performance. Here, Column 2 uses the same colorbar as Column 1 for consistent

Table 5: Mean LNCC (Std. Dev.) comparison ($N = 1,500$) on the LMSB harmonization results using the INNs trained from w/o Volume Preservation, w/o Data Augmentation, and LMSB-INN (Ours), respectively. Examples of the harmonized images are shown in Fig. 8. The LNCCs were calculated between the harmonized images and the original input images. Higher LNCCs indicate a better preservation of the anatomical structure. Bold numbers indicate the best result in that row. Asterisks indicate statistical significance (i.e., paired t-test comparing 1st and 2nd best results gave p-value $< 0.05$).

|  |  | w/o Volume Preservation | | w/o Data Augmentation | | LMSB-INN (Ours) | |
|---|---|---|---|---|---|---|---|
|  |  | SDE | ODE | SDE | ODE | SDE | ODE |
| LNCC | **Spectralis to Cirrus** | 0.045 (0.004) | 0.234 (0.063) | 0.147 (0.013) | 0.518 (0.042) | 0.165 (0.016) | **0.662**$^*$ (0.052) |
|  | **Cirrus to Spectralis** | 0.037 (0.011) | 0.239 (0.083) | 0.139 (0.014) | 0.510 (0.044) | 0.164 (0.019) | **0.692**$^*$ (0.030) |

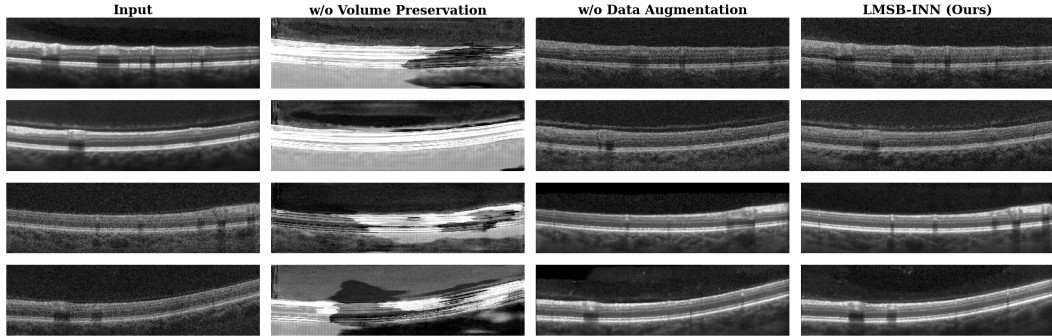

Figure 8: Anatomy preservation of LMSB harmonization with the ODE sampling strategy. Column 1 shows the original input images before harmonization, the same as Column 1 in Fig. 6. Rows 1–2 show two examples of Spectralis to Cirrus harmonization, and Rows 3–4 show two examples of Cirrus to Spectralis harmonization. We trained a LMSB for each INN that was trained from w/o Volume Preservation, w/o Data Augmentation, and LMSB-INN (Ours). We used the same training parameters for LMSB, which we described in Network Training. Columns 2–4 show the harmonized images of the LMSBs trained from w/o Volume Preservation, w/o Data Augmentation, and LMSB-INN (Ours), respectively.

comparison. From the results in Column 3, we see that without data augmentation, although most retinal layer structures are preserved before and after harmonization, the detailed anatomical structures that are not in the segmentation label maps are changed, such as vessels and shadows. The results of LMSB with both volume preservation and data augmentation are shown in Column 4, where we see that not only the retinal layers but also the vessels, shadows, and choroids are well preserved before and after harmonization.

## C Additional Qualitative Results

We demonstrate the harmonization quality of our proposed method LMSB with ODE sampling on several examples, with Spectralis to Cirrus harmonization in Fig. 9 and Cirrus to Spectralis harmonization in Fig. 10. From these qualitative results, we see that the retinal anatomy before and after harmonization is well preserved, including anatomical structures such as retinal layers, vessels, shadows, and choroids. We note that some of these structures, such as vessels, shadows, and choroids, do not appear in the segmentation label maps that were used to train the INN. Moreover, from the Cirrus to Spectralis harmonization results in Fig. 10, we see that many anatomical structures such as vessels, shadows, and choroids that are difficult to visualize in the original Cirrus B-scan become clearer in the harmonized B-scan with the Spectralis contrast. This suggests that the proposed harmonization method, LMSB, may benefit downstream tasks such as vessel segmentation.

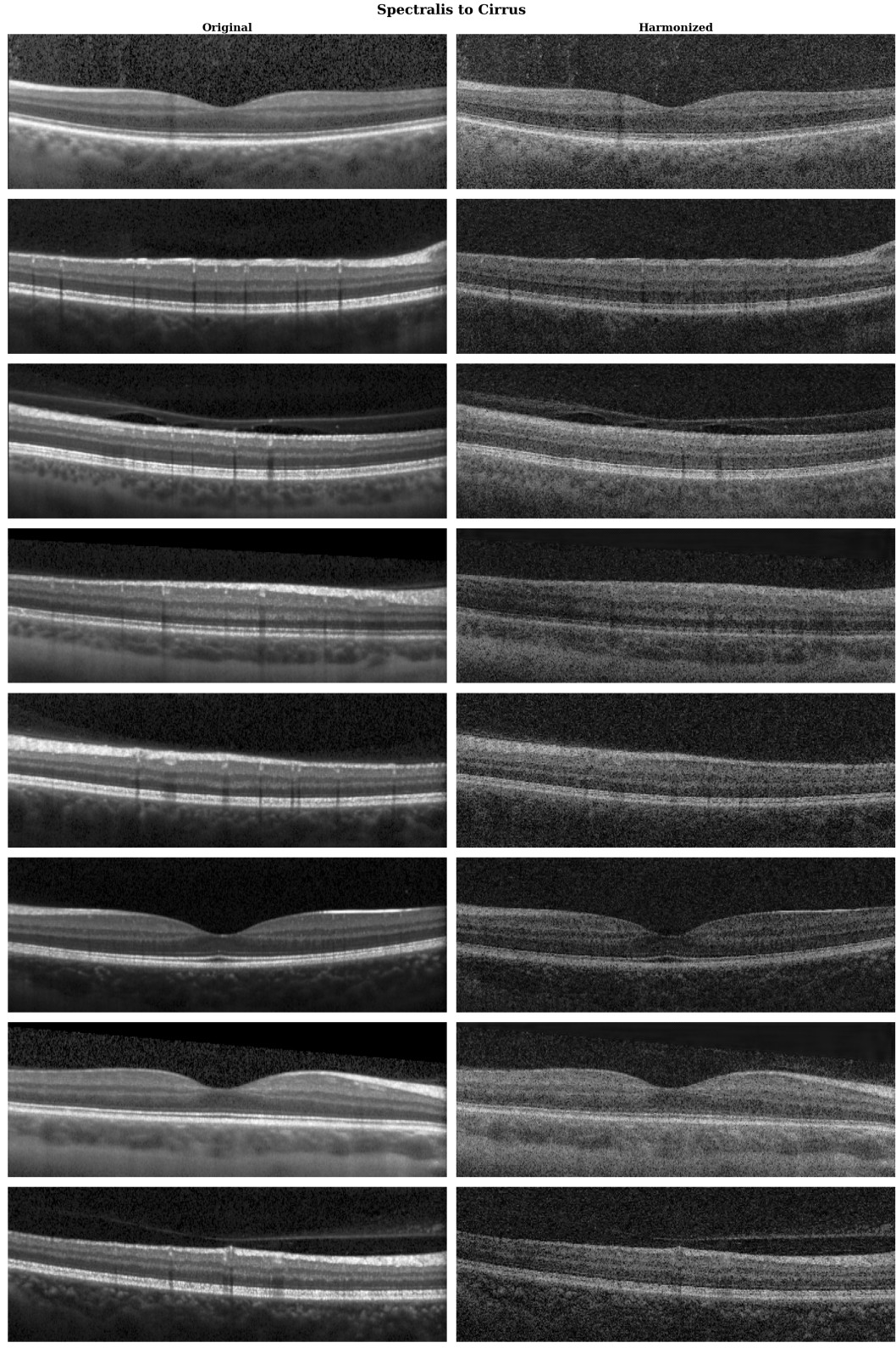

Figure 9: Random examples of Spectralis to Cirrus harmonization by LMSB using the ODE sampling strategy. The left column shows the original Spectralis OCT B-scans. The right column shows the synthetic Cirrus OCT B-scans that are harmonized from the left column.

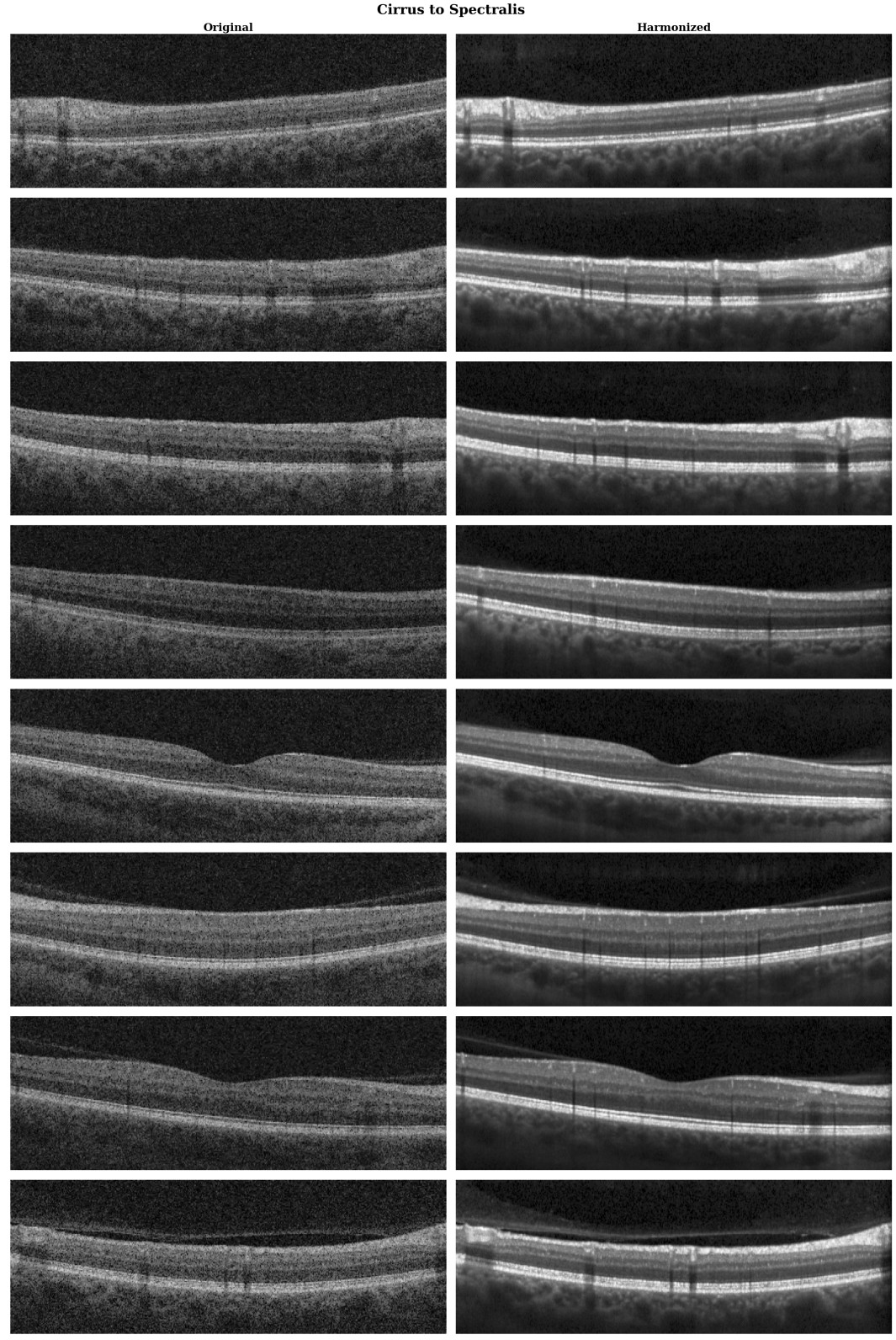

Figure 10: Random exmaples of Cirrus to Spectralis harmonization by LMSB using the ODE sampling strategy. The left column shows the original Cirrus OCT B-scans. The right column shows the synthetic Spectralis OCT B-scans that are harmonized from the left column.

Table 6: FID comparison for both Spectralis to Cirrus and Cirrus to Spectralis harmonization. Before harmonization, the FID between the $1,500$ Cirrus B-scans and the $1,500$ Spectralis B-scans in the testing dataset is 140.53. After harmonization, the FIDs were calculated between the $1,500$ synthetic B-scans and the $1,500$ original B-scans with the target contrast. Bold numbers indicate the best result for that subtask.

| | Spectralis to Cirrus | | | | | Cirrus to Spectralis | | | | |
| | DDIB | DSBM | | LMSB | | DDIB | DSBM | | LMSB | |
| | ODE | SDE | ODE | SDE | ODE | ODE | SDE | ODE | SDE | ODE |
| FID | 48.48 | 18.48 | 24.26 | **17.24** | 25.34 | 50.97 | 34.99 | **31.39** | 41.72 | 32.97 |

# D   Additional Quantitative Results

## D.1   Domain Consistency Comparison

To measure the domain consistency, we calculated Fréchet inception distance (FID) before and after harmonization with respect to the testing dataset. Before harmonization, the FID between the $1,500$ Cirrus B-scans and the $1,500$ Spectralis B-scans in the testing dataset is 140.53. We conducted both Spectralis to Cirrus harmonization and Cirrus to Spectralis harmonization. After harmonization, the FIDs were calculated between the $1,500$ synthetic B-scans and the $1,500$ original B-scans with the target contrast. The FID results are shown in Table 6. We found that FID improves after harmonization with respect to the target contrast for all the harmonization methods. Moreover, by comparing the FIDs of different methods after harmonization, we can see that both DSBM and LMSB have better FID than DDIB. LMSB has slightly larger FID than DSBM, which is expected because the LMSB does not directly apply harmonization between two original data domains.

## D.2   Reducing Inter-Device Measurement Bias

To evaluate the impact of the LMSB on this inter-device bias, we performed a downstream segmentation and retinal layer thickness measurement on 115 pairs of Cirrus and Spectralis volumes collected from 59 subjects and 115 eyes, which is separate from the training and testing dataset used in this paper. These 115 pairs of scans were completed contemporaneously. The retinal layer thickness measurements from these two devices disagree in general, which is an important clinical issue in OCT imaging when comparing the measurements from different devices on monitoring a subject that is scanned on different machines during the time course of the study.

We conducted the following harmonization experiments using LMSB with ODE sampling strategy on the paired OCT dataset: 1) Spectralis to Cirrus harmonization; and 2) Cirrus to Spectralis harmonization. In the first experiment, we harmonize the 115 Spectralis volumes to the Cirrus scanner. We computed the retinal layer thickness measurement differences between synthetic Cirrus (synC) volumes and corresponding original Cirrus (orgC) volumes, and we compared them with the differences between original Spectralis (orgS) volumes and original Cirrus (orgC) volumes. The retinal layer thickness measurements were done by first applying a deep learning segmentation algorithm and then computing the average layer thickness in a $5 \times 5$ $mm^2$ square centered on the fovea. The second experiment is similar to the first with the exception that we harmonize the 115 Cirrus volumes to the Spectralis scanner. We report difference in thickness measurements between the synthetic Spectralis (synS) volumes and corresponding original Spectralis (orgS) volumes, as well as the differences between original Cirrus (orgC) volumes and original Spectralis (orgS) volumes. The results are shown in Table 7, with the mean and the standard deviation of the signed difference of the retinal layer thickness measurement.

For both results, we see that harmonization helps reduce the inter-device bias for all retinal layers including the most important retinal layer GCIPL. Moreover, paired t-tests show statistical significance (p-value $< 0.05$) for most cases. We need to highlight that the proposed method was originally developed such that the thickness measurements from original volumes and harmonized volumes are consistent. However, from the results in Table 1, we see that we still do not obtain the exact anatomy consistency with the proposed method. From these additional experiments on the paired dataset, we see that this remaining anatomy inconsistency actually helps reduce the inter-device bias.

Table 7: Mean difference (Std. Dev.) comparison ($N = 115$) in units of $\mu m$ for thickness measurements across eight retinal layers before and after harmonization for both Spectralis to Cirrus and Cirrus to Spectralis harmonization. Before harmonization, the thickness measurement differences were derived from the two original OCT volumes. After harmonization, the thickness measurement differences were derived from the synthetic and corresponding original OCT volumes. Bold numbers indicate reduced inter-device thickness measurement bias in that row for that subtask. Asterisks indicate statistical significance (i.e., paired t-test comparing before and after harmonization gave p-value $< 0.05$). **Key:** RNFL: retinal nerve fiber layer; GCIPL: ganglion cell and inner plexiform layer; INL: inner nuclear layer; OPL: outer plexiform layer; ONL: outer nuclear layer; IS: inner segment; OS: outer segment; RPE: retinal pigment epithelium complex; orgC: original Cirrus; orgS: original Spectralis; synC: synthetic Cirrus; synS: synthetic Spectralis.

| | **Spectralis to Cirrus** | | **Cirrus to Spectralis** | |
| | orgS $-$ orgC | synC $-$ orgC | orgC $-$ orgS | synS $-$ orgS |
| --- | --- | --- | --- | --- |
| RNFL | 9.66 (2.70) | **5.44**$^*$ (5.35) | $-9.66$ (2.70) | $-$**3.23**$^*$ (3.37) |
| GCIPL | $-4.75$ (1.31) | $-$**1.89**$^*$ (2.26) | 4.75 (1.31) | **2.66**$^*$ (1.92) |
| INL | $-2.64$ (1.19) | $-$**1.79**$^*$ (1.92) | 2.64 (1.19) | **1.09**$^*$ (1.57) |
| OPL | 0.63 (0.66) | $-$**0.16**$^*$ (1.38) | $-0.63$ (0.66) | **0.26**$^*$ (0.75) |
| ONL | $-4.20$ (1.68) | $-$**2.33**$^*$ (2.64) | 4.20 (1.68) | **0.48**$^*$ (2.35) |
| IS | $-0.31$ (0.41) | $-$**0.06**$^*$ (1.04) | 0.31 (0.41) | **0.24** (0.48) |
| OS | 1.16 (1.04) | $-$**0.18**$^*$ (1.27) | $-1.16$ (1.04) | **0.38**$^*$ (0.81) |
| RPE | 1.64 (1.23) | $-$**0.27**$^*$ (0.98) | $-1.64$ (1.23) | **0.74**$^*$ (1.26) |

## E   Computational Resources

The proposed model LMSB requires only a single GPU with 48 GB of memory for training and 15 GB for evaluation when the batch size is 16. However, in practice we used several different GPUs in parallel to train and test different models with different hyperparameters. To train all the models including the proposed model, comparison models and ablation models, we used seven GPUs in total, including four NVIDIA A40 (48 GB), two NVIDIA RTX A6000 (48 GB), and one QUADRO RTX 8000 (48 GB).

