# OpenReview forum: "Optical Coherence Tomography Harmonization with Anatomy-Guided Latent Metric Schrödinger Bridges"
_NeurIPS.cc/2025/Conference — NeurIPS 2025 poster_

### Official Review · Reviewer_ymJv · 2025-06-27

**Clarity:** 2
**Significance:** 3
**Originality:** 2
**Rating:** 4
**Confidence:** 1

**Summary:**

The paper introduces a new method, the latent metric Schrödinger bridge (LMSB), which improves anatomical consistency for the harmonisation of medical images. They train an invertible network that maps the OCT images into a latent Euclidean space where distances between images with the same anatomical structure are minimized, The authors show that the method was able to harmonise OCT images.

**Questions:**

1. Could the authors elaborate on how the method would apply to other medical images? The paper discusses the case of OCT images. Would there be any problems in applying it to MRI or Ultrasounds? Would the variability of the acquisition of ultrasound make it harder to apply the proposed method ? In the limitations section, the authors mention that the complexity of the anatomy might add complexity to the method. Can the authors elaborate on that point?
2. Could the authors provide a more thorough statistical analysis of the quantitative results presented?
3. Can the authors expand on what computational resources were used in this work?
4. Line 54: please elaborate which assumptions from OT and SB are not correct for medical image harmonization.
5. The authors report the MAE as the evaluation metric, could the authors clarify why this is the only metric reported instead of FID and more traditional generative metrics?

**Ethical Concerns:**

["NO or VERY MINOR ethics concerns only"]

**Final Justification:**

Thanks for the response. I think it's important to include the FID results in the paper, and it's also good to hear that you'll be adding a discussion on the generalizability of the pre-processing steps in the final version. That said, I'll keep my score as is.

**Limitations:**

yes

**Quality:**

3

**Strengths And Weaknesses:**

Strengths:
- Comparison to other methods
- The paper tackles the challenge of domain shift caused by different scanners or acquisition parameters

Weaknesses:
- There is a lack of statistical significance that supports the main claim of the paper.

---

> ### Author Rebuttal · Authors · 2025-07-31
>
> We thank the reviewer for the constructive comments that helps improve the quality of our paper. We address the concerns as follows.
>
> **1. Statistical analysis.**
>
> Following the reviewer’s suggestion, we conducted a statistical analysis by one-side paired T-test to see if the best mean value (the bold number) is statistically better than the second best mean value, and the test results show a statistical significance for most cases in both Table 1 and Table 2.
>
> Note that in the original paper, we have reported the variance of each metric in both Table 1 and Table 2, and our test dataset is relatively large (N=1,500 B-scans), which already shows some level of statistical significance. We will add the additional one-side paired T-test results to the final paper.
>
> **2. Generalization to other medical images, such as MRI and ultrasounds.**
>
> Theoretically LMSB can generalize to other imaging modalities. Because the training of LMSB only requires unpaired datasets and segmentation label maps associated with the data. Unpaired datasets are easier to obtain than paired datasets in clinical settings for both MRI and ultrasound, and the segmentation label maps are theoretically available if we train segmentation models on both datasets.
>
> **3. Would the variability of the acquisition of ultrasound make it harder to apply the proposed method?**
>
> The variability of the acquisition of ultrasound could potentially complicate the pre-processing steps but should not affect the application of the proposed method. Currently, we have two pre-processing steps for OCT, including a crop and a resize, to have both OCT B-scans from two devices of the same image size, digital resolution, and field of view. With ultrasound, the acquisition could be much more complex, and the dataset may contain images with quite different image sizes, image resolutions, fields of view, etc, which complicates the pre-processing steps. However, once these pre-processing steps are solved, the application of the proposed method should be the same. We will add a discussion on the generalizability of the pre-processing steps to the final paper.
>
> **4. How does complexity of anatomy affect the proposed method?**
>
> The complexity of the anatomy would affect the proposed method. As we discussed in Fig. 1 in the original paper, we can either compress the distance along the contrast direction or stretch the distance along the anatomy direction to reduce the anatomical shift. We did compress the distance along the contrast direction by training an invertible neural network, but we didn’t optimize the distance along the anatomy direction. The current segmentation label map assigns unique numbers to different retinal layers, and the number is increasing from the top layer to the bottom layer. This is not necessarily optimal. For example, if we have more layers, then the contrast between adjacent layers in the segmentation label map becomes lower, and the same anatomy shift would lead to a smaller Euclidean distance, which effectively compresses the distance along the anatomy direction and thus will affect the performance of the proposed method. We will add elaborate on the corresponding discussion in the final paper.
>
> **5. Computational resources.**
>
> The proposed model required only a single GPU with 48GB of memory for training and 15GB for evaluation when the batch size is 16. However, in practice we used several different GPUs in parallel to train and test different models with different hyperparameters. To train all the models including the proposed model, comparison models and ablation models, we used seven GPUs in total, including four NVIDIA A40 (48GB), two NVIDIA RTX A6000 (48GB), and one QUADRO RTX 8000 (48GB). We will add these details about the computational resources we used to the final paper.
>
> **6. Assumption from OT and SB for medical image harmonization.**
>
> The assumption behind OTs and SBs is that images with the same anatomy and different contrast should have small Euclidean distances. This is because the OTs and SBs are trying to find an optimal coupling that minimizes the overall transport cost, where the transport cost is the square of Euclidean distance between images. This assumption is not correct because images with the same anatomy but different contrast do not necessarily have small Euclidean distance, and thus OTs and SBs may not find a correct correspondence between them. We have discussed this in detail in “issues of Euclidean metrics and potential solutions” in the original paper, but we will add a brief explanation around line 54 in the final paper.
>
> **7. Traditional generative metrics such as FID.**
>
> We agree with the reviewer that traditional generative metrics such as FID are important. We use MAE and LNCC to measure anatomical consistency because improving the anatomical consistency is the goal of the paper. To measure the domain consistency, we calculated Fréchet inception distance (FID) before and after harmonization with respect to the testing dataset. We found that FID improves after harmonization with respect to the target contrast. Moreover, we compare the FIDs of different methods after harmonization as shown in the following table, where we can see that both DSBM and LMSB have better FID than DDIB. LMSB has slightly larger FID than DSBM, which is expected because the LMSB does not directly apply harmonization between two original data domains. We will add this FID evaluation to the final paper.
>
> | | |Spectralis|to|Cirrus| | |Cirrus|to|Spectralis| |
> |:-:|:-:|:-:|:-:|:-:|:-:|:-:|:-:|:-:|:-:|:-:|
> | |DDIB|DSBM|DSBM|LMSB|LMSB|DDIB|DSBM|DSBM|LMSB|LMSB|
> | |ODE|SDE|ODE|SDE|ODE|ODE|SDE|ODE|SDE|ODE|
> |Before|140.53|140.53|140.53|140.53|140.53|140.53|140.53|140.53|140.53|140.53|
> |After|48.48|18.48|24.26|**17.24**|25.34|50.97|34.99|**31.39**|41.72|32.97|

---

> > ### Comment · Reviewer_ymJv · 2025-08-06
> >
> > Thanks for the response. I think it's important to include the FID results in the paper, and it's also good to hear that you'll be adding a discussion on the generalizability of the pre-processing steps in the final version. That said, I'll keep my score as is.

---

> ### Author Response · Authors · 2025-08-06
>
> Thank you! We will include them as you suggested.

---

### Official Review · Reviewer_tKW4 · 2025-07-01

**Clarity:** 3
**Significance:** 3
**Originality:** 4
**Rating:** 4
**Confidence:** 4

**Summary:**

This paper develops a method, namely anatomy-guided latent metric Schrödinger bridge (LMSB), for OCT image harmonization to reduce anatomical shifts. The authors provide theoretical, quantitative, and qualitative analysis of the proposed algorithm on a private dataset.

**Questions:**

1. Will the pre-processing of the images affect the performance of the algorithm? What is the original size of the OCT images?
2. The authors may consider adding more evaluation metrics, such as PSNR, SSIM or some Edge-preserving metrics, to better demonstrate the performance of the algorithm. Even though the pairs are not pixel-wise matched, the authors may consider using registration algorithms before calculating these metrics.
3. The authors may consider collecting feedback from the experts to validate the performance of the algorithm.

**Ethical Concerns:**

["NO or VERY MINOR ethics concerns only"]

**Limitations:**

The authors may consider discussing the limitations of the societal impacts, not just the technical impacts.

**Quality:**

3

**Strengths And Weaknesses:**

Strengths:
1. The proposed LMSB framework enhances anatomical consistency in medical image harmonization by mapping images to a learned latent Euclidean space using an invertible neural network.
2. The authors provide a rigorous theoretical analysis of why conventional Euclidean transport costs in SB and optimal transport are unsuitable for medical images.

Weakness:
1. The method is only demonstrated on retinal OCT images, which have relatively simple and regular anatomical structures.
2. There is no evidence that LMSB can generalize to other imaging modalities, or even for other OCT vendors. For example, the two OCT systems in the paper (Zeiss Cirrus OCT and Heidelberg Spectralis OCT ) are SDOCT systems. Will the algorithm be effective between SDOCT and SSOCT?
3. The authors claim that the proposed algorithm is for the image harmonization task. However, the evaluation metrics include MAE and LNCC. Can these two metrics measure the domain consistency between two modalities?

---

> ### Author Rebuttal · Authors · 2025-07-31
>
> We thank the reviewer for the thorough list of the strengths and constructive comments about the weaknesses of our paper. We address the concerns as follows.
>
> **1. The proposed method is only demonstrated on retinal OCT images.**
>
> We agree with the reviewer that this is a limitation of the paper and we pointed out this limitation in the discussion. Our application was to retinal OCT because that is what our group is interested in. However, the proposed method is a general framework and can generalize to other modalities. Please see the following response for the generalizability of the proposed method.
>
> **2. Can the proposed method generalize to other imaging modality or other OCT vendors?**
>
> Theoretically yes, LMSB can generalize to other imaging modalities or other OCT vendors. Currently the training of LMSB only requires unpaired datasets and segmentation label maps associate with the data. The unpaired datasets are easier to obtain than paired datasets in clinical settings, and the segmentation label maps are theoretically available if we train segmentation models on both datasets.
>
> **3. Can MAE and LNCC measure domain consistency between two modalities?**
>
> We agree with the reviewer that measuring the domain consistency is an important metric. MAE and LNCC however measures the anatomical consistency instead of domain consistency. We use these two metrics because improving the anatomical consistency is the goal of the paper. To measure the domain consistency, we calculated Fréchet inception distance (FID) before and after harmonization with respect to the testing dataset. We found that FID improves after harmonization with respect to the target contrast. Moreover, we compare the FIDs of different methods after harmonization as shown in the following table, where we can see that both DSBM and LMSB have better FID than DDIB. LMSB has slightly larger FID than DSBM, which is expected because the LMSB does not directly apply harmonization between two original data domains. We will add this FID evaluation to the final paper.
>
> | | |Spectralis|to|Cirrus| | |Cirrus|to|Spectralis| |
> |:-:|:-:|:-:|:-:|:-:|:-:|:-:|:-:|:-:|:-:|:-:|
> | |DDIB|DSBM|DSBM|LMSB|LMSB|DDIB|DSBM|DSBM|LMSB|LMSB|
> | |ODE|SDE|ODE|SDE|ODE|ODE|SDE|ODE|SDE|ODE|
> |Before|140.53|140.53|140.53|140.53|140.53|140.53|140.53|140.53|140.53|140.53|
> |After|48.48|18.48|24.26|**17.24**|25.34|50.97|34.99|**31.39**|41.72|32.97|
>
>
> **4. Will the pre-processing affect the algorithm? What is the original size of OCT images?**
>
> We agree with the reviewer’s concern over the impact of pre-processing. For this reason, the preprocessing techniques we chose were minimal: only cropping and resizing to have both Cirrus and Spectralis B-scans of the same image size, digital resolution, and field of view. We do not perform any other processing steps that could affect the image intensities, such as histogram equalization or bias field correction. The intent was exactly to address the reviewer’s concern that excessive preprocessing would affect the domain appearance of each scan.
>
> The original size of Cirrus B-scans is 1024x512 (axial x lateral) and the original size of Spectralis B-scans is 496x1024 (axial x lateral). The axial resolution of Cirrus B-scans is 2 um/pixel, and the axial resolution of Spectralis B-scans is 4 um/pixel. Both B-scans cover a 6 mm lateral scanning range. We crop the Cirrus B-scans axially to 512x512 and resize them to 128x512, and we crop the Spectralis B-scans axially to 256x1024 and resize them to 128x512, such that they have the same image size, digital resolution, and field of view. We will add these details to the final paper.
>
> **5. Adding alternative metrics such as PNSR SSIM for paired B-scans after registration.**
>
> We agree with the reviewer that paired B-scans would lead to improved validation. Reviewer ZM1d also brought up a similar concern. However, creating paired B-scans using registration as described by the reviewer is not straightforward. This is due to the following three aspects: 1) Deformable registration is required due to the difference of optical setups and post alignment methods between the two devices. The retinal curvature between two different scanners is different within the same scanning region, and thus there is no affine relationship between volumes acquired from the two different scanners. 2) Registering highly anisotropic volumes is inaccurate and would hinder validation. For example, in our dataset, Cirrus volumes contain 128 B-scans, each with 512 A-scans, but Spectralis volumes contain 49 B-scans, each with 1024 A-scans. Even with affine registration, this kind of anisotropy would lead to egregious aliasing artifacts that would corrupt analysis. 3) Field bias differs between the two scanners. Spectralis volumes often have abrupt field variations, i.e., the mean intensity may change abruptly between adjacent B-scans, in which case an aligned B-scan after registration may not look like a Spectralis B-scan. This adds to the difficulty of creating real paired B-scans.
>
> However, we do plan to continue work to improve validation of this technique for OCT volumes. Additional evaluations are out-of-scope for this work, but we will mention this limitation in our discussion and discuss how future work can investigate and evaluate the method further.
>
> **6. Feedback from experts.**
>
> We agree with the reviewer that feedback from experts is important. We have extensive medical expertise in our team, specifically we have two medical doctors with 55 years of experience between them. They have provided extensive feedback throughout the development and testing processing of the proposed work.

---

### Official Review · Reviewer_rEQb · 2025-07-03

**Clarity:** 2
**Significance:** 2
**Originality:** 3
**Rating:** 4
**Confidence:** 3

**Summary:**

Existing image harmonization approaches distort the underlying anatomy (often severely in my experience.) This approach uses the existing annotations of anatomy to suppress this issue in the framework of a Diffusion Schrödinger Bridge. The result is a meaningful improvement in faithfulness.

**Questions:**

The paper states that the dataset consisted of 16500 slices from each machine, coming from ~330 scans, and split into train and test set at a ration of 15000 slices to 1500 slices. Please confirm whether the train-test split was done at the scan level or the slice level- the phrasing makes it sound concerningly like the split was done at the slice level, which would introduce heavy test set contamination.

**Ethical Concerns:**

["NO or VERY MINOR ethics concerns only"]

**Limitations:**

yes

**Paper Formatting Concerns:**

No formatting concerns

**Quality:**

2

**Strengths And Weaknesses:**

There is a great deal of math going on, but digging to the root of it it appears that the latent variable z is just the segmentations associated with the data, and so the correspondence that is being preserved is just that the segmentation shouldn't change. This is admirable and of course the goal, but it could have been stated more clearly. The specific way of using segmentations as a flow matching target is clever and well integrated to the schroedinger bridge framework.

On the other side of clarity, Figure 1 is excellently done, and was central to my ability to grok the paper's mathematical content.

---

> ### Author Rebuttal · Authors · 2025-07-31
>
> We thank the reviewer for the commendation. We are glad to hear that Fig. 1 helps improve the clarity of the mathematical content. We address the concerns and clarify some misunderstandings as follows.
>
> **1. The correspondence that is being preserved is just that the segmentation shouldn’t change.**
>
> Although this is almost true, we want to highlight that both Cirrus and Spectralis B-scans are not perfectly mapped to the segmentation label maps, which can be seen from Fig. 4. The harmonization then applies to the latent representation of the original images through the invertible neural network, which effectively reduces the change of segmentation but not necessarily eliminate it. Therefore, we cannot say that the segmentation does not change.
>
> **2. The core idea could have been stated more clearly.**
>
> Following the previous response, we plan to give an intuitive explanation of the core idea as follows: Intuitively, the invertible neural network brings both Cirrus and Spectralis B-scans closer to a common latent contrast while preserving the anatomy using segmentation label maps so that the anatomy shift, i.e., segmentation change, is reduced during harmonization. We will add this intuitive explanation to the final paper.
>
> **3. Test set contamination.**
>
> We split the training and testing datasets by volumes so there is no data leakage. Reviewer ZM1d also brought up a similar concern. For clarity, we will modify the following details in the final paper.
>
> The dataset consists of 388 Cirrus volumes and 338 Spectralis volumes. The 388 Cirrus volumes come from 194 subjects (388 eyes in total). The 338 Spectralis volumes come from 165 subjects (269 eyes in total). Note that there are repeated scans on the same eye for Spectralis. Training and testing splits are done by subject: 352 Cirrus volumes from 176 subjects (352 eyes) and 307 Spectralis volumes from 156 subjects (252 eyes) for training and 36 Cirrus volumes from 18 subjects (36 eyes) and 31 Spectralis volumes from 9 subjects (17 eyes) for testing. Our experiments are in 2D and operate on B-scans independently. Because each Cirrus volume contains 128 B-scans and each Spectralis OCT volume contains 49 B-scans over the same field of view 6x6 mm2, the Cirrus volumes have denser B-scan sampling. Therefore, to reduce anatomical redundancy across individual Cirrus volume B-scans, we extract every third B-scan. Specifically, in the training dataset, we extract 15,000 B-scans in the training dataset and 1,500 B-scans in the testing dataset for both Cirrus and Spectralis, resulting in 30,000 training B-scans and 3,000 testing B-scans with no subject data leakage between train and test splits.

---

### Official Review · Reviewer_ZM1d · 2025-07-03

**Clarity:** 3
**Significance:** 3
**Originality:** 3
**Rating:** 4
**Confidence:** 5

**Summary:**

This paper tackles the long-standing “anatomy-shift” problem that occurs when applying Schrödinger Bridge (SB) or entropy-regularized optimal transport directly to medical images. In this manuscript, the authors proposed anatomy guided latent metric Schrödinger Bridge (LMSB) to achieve optical coherence tomography (OCT) image harmonization. The authors demonstrated that directly using SB might cause anatomy shift, which could affect the downstream tasks. The main technical contributions is mapping OCT images to a latent Euclidean metric space via anatomy guidance, which was demonstrated to have better performance than DDIB and DSBM.

**Questions:**

1. Why did authors choose segmentation label maps as the sole anatomy-preserving coordinate? Have them explored alternative learned metrics.
2. Can author address my concern on the dataset preparation?
3. Can author acquire scans of the same retinal region in a single participant with both OCT devices and display the harmonized output and the true target image side by side?

**Ethical Concerns:**

["NO or VERY MINOR ethics concerns only"]

**Final Justification:**

The authors have addressed most of my concerns except registering two volumes from different devices, which might be difficult to complete within a few weeks. Hence, I changed the score from 3 to 4.

**Limitations:**

Yes

**Quality:**

3

**Strengths And Weaknesses:**

Strengths:

1. The paper formalizes how pixel-space Euclidean costs confound contrast variation with anatomical differences.
2. Training a network to “flatten” anatomy into a Euclidean metric, followed by SB in that space, is conceptually elegant and empirically effective.
3. The proposed method does not require paired data, which would be very helpful in the field of medical imaging.
4. Compared to other pipelines, the proposed method displayed expecptional improved. And there's almost no anatomy shift for images generated by LMSB ODE.

Weaknesses:

1. I have a concern about the dataset. For the dataset preparation, the authors acquired 16, 500 Cirrus B-scans sampled from 387 Cirrus OCT volumes, and 16, 500 Spectralis B-scans sampled from 337 Spectralis OCT volumes. And then, they split the B-scans into 15,000 and 1,500 for each OCT device. The field of view of OCT is quite small (usually 3x3, 6x6 or 12x12 mm2) compared to other imaging modalities like ultrasound and CT. Consequently, the B-scans within a single volume are often highly similar, especially those that are adjacent or otherwise close to one another. Hence, I feel like splitting the dataset by B-scans rather than the volumes is a kind of data leakage and cannot reflect the performance in the 'real world' setting.
2. The current validation is insufficient. I suggest acquiring scans of the same retinal region in a single participant with both OCT devices and then registering the two volumes (both two devices mentioned in the manuscript can automately take OCT images located the central fovea of the retina, so the subsquent registration is not quite hard). This setup would enable a direct, voxel-wise comparison between the synthesized and ground-truth images, clearly revealing any discrepancies.
3. This works relies on deep learning segementation of the retina layers, so any bias or error in the segmentation could propagate and adversely affect the subsequent LMSB stage.
4. For the inner retina, the current OCT/ophthalmology studies focus a lot on RNFL-GCL as it's related to glucoma (biomaker: thickness of the layer) or other nuerodegenration diseases (biomarker: morphology of the retina ganglion axon bundles). Unfortunately, the proposed method does not deliver satisfactory performance on this critical layer.

---

> ### Author Rebuttal · Authors · 2025-07-31
>
> We thank the reviewer for the thorough list of the strengths and constructive comments about the weaknesses of our paper. We address the concerns as follows.
>
> **1. Training and testing data leakage.**
>
> We split the training and testing datasets by volumes so there is no data leakage. Reviewer rEQb also brought up a similar concern. For clarity, we will modify the following details in the final paper.
>
> The dataset consists of 388 Cirrus volumes and 338 Spectralis volumes. The 388 Cirrus volumes come from 194 subjects (388 eyes in total). The 338 Spectralis volumes come from 165 subjects (269 eyes in total). Note that there are repeated scans on the same eye for Spectralis. Training and testing splits are done by subject: 352 Cirrus volumes from 176 subjects (352 eyes) and 307 Spectralis volumes from 156 subjects (252 eyes) for training and 36 Cirrus volumes from 18 subjects (36 eyes) and 31 Spectralis volumes from 9 subjects (17 eyes) for testing. Our experiments are in 2D and operate on B-scans independently. Because each Cirrus volume contains 128 B-scans and each Spectralis OCT volume contains 49 B-scans over the same field of view 6x6 mm2, the Cirrus volumes have denser B-scan sampling. Therefore, to reduce anatomical redundancy across individual Cirrus volume B-scans, we extract every third B-scan. Specifically, in the training dataset, we extract 15,000 B-scans in the training dataset and 1,500 B-scans in the testing dataset for both Cirrus and Spectralis, resulting in 30,000 training B-scans and 3,000 testing B-scans with no subject data leakage between train and test splits.
>
> **2. Validation on paired B-scans.**
>
> We agree with the reviewer that paired B-scans would lead to improved validation. Reviewer tKW4 also brought up a similar concern. However, creating paired B-scans using registration as described by the reviewer is not straightforward. This is due to the following three aspects: 1) Deformable registration is required due to the difference of optical setups and post alignment methods between the two devices. The retinal curvature between two different scanners is different within the same scanning region, and thus there is no affine relationship between volumes acquired from the two different scanners. 2) Registering highly anisotropic volumes is inaccurate and would hinder validation. For example, in our dataset, Cirrus volumes contain 128 B-scans, each with 512 A-scans, but Spectralis volumes contain 49 B-scans, each with 1024 A-scans. Even with affine registration, this kind of anisotropy would lead to egregious aliasing artifacts that would corrupt analysis. 3) Field bias differs between the two scanners. Spectralis volumes often have abrupt field variations, i.e., the mean intensity may change abruptly between adjacent B-scans, in which case an aligned B-scan after registration may not look like a Spectralis B-scan. This adds to the difficulty of creating real paired B-scans.
>
> However, we do plan to continue work to improve validation of this technique for OCT volumes. Additional evaluations are out-of-scope for this work, but we will mention this limitation in our discussion and discuss how future work can investigate and evaluate the method further.
>
> **3. Segmentation bias or error may affect harmonization.**
>
> A consistent segmentation bias for both Cirrus and Spectralis B-scans does not affect the subsequent harmonization, and an accurate segmentation is not required for training the invertible neural network.
>
> The first statement can be seen from Eq. 5, where the only requirement for the latent variable z is that it should be the same for images with the same anatomy but different contrast. Therefore, a consistent segmentation bias for both Cirrus and Spectralis B-scans does not affect the subsequent harmonization. The second statement can be seen from Fig. 4, where we can see that both the Cirrus and Spectralis B-scans are not perfectly mapped to the segmentation label maps and some details such as vessel and shadows are preserved. This suggests that an accurate segmentation is not required. Both have been discussed in the original paper.
>
> However, a different segmentation bias between Cirrus and Spectralis B-scans will affect the subsequent LMSB harmonization because it violates the assumption in Eq. 5, for example, if a Cirrus segmentation is biased upwards and a Spectralis segmentation is biased downwards. We will add this as a limitation of our method to the discussion section.
>
> **4. Performance on RNFL-GCL and application to ophthalmology.**
>
> We agree with the reviewer that RNFL-GCL is important in many applications in ophthalmology but emphasize that our paper is mainly a methodology development and evaluation that tries to improve anatomical consistency in harmonization instead of applying it to a specific application. As an aside, in our group we are also interested in some potential important applications other than RNFL-GCL, including hyper-reflective foci (HRF) detection and GCL-IPL separation. HRFs refer to small bright spots seen in OCT in different retinal layers that do not cast shadows, and it is recognized as a biomarker of retinal degeneration. GCL and IPL are often grouped together in retinal OCT segmentation because they have subtle differences in tissue properties, and the independent segmentation of GCL and IPL would introduce a more important biomarker from GCL alone in ophthalmology. For Cirrus B-scans, both HRF and the GCL-IPL boundary are ambiguous because of speckle and noise. In some of the results of Cirrus to Spectralis harmonization in Fig. 10 in the supplemental material, we observe that both HRF and the GCL-IPL separation becomes more visible in the synthetic Spetralis B-scans, which may benefit HRF detection and GCL-IPL segmentation from the original Cirrus B-scans. We will add these observations when presenting the results in Fig. 10. Moreover, the other comparison methods also do not perform well on RNFL-GCL boundary compared to other retinal boundaries, which is interesting and worthy of further investigation. We will add this discussion and observation when presenting the results in Table 1.
>
> **5. Reasons for choosing segmentation label map and alternative learned metrics.**
>
> The reason for choosing segmentation label map is that it is a sufficient and straightforward design that satisfies the condition of Eq. 5, i.e., the latent variable z should be the same for images with the same anatomy but different contrast. Moreover, this choice is easy to generalize to other modalities and anatomies because the proposed method is potentially applicable when segmentation label maps exist.
>
> Alternative learned metrics include variants of the segmentation label maps. The current segmentation label map assigns unique numbers to different retinal layers, and the number is increasing from the top layer to the bottom layer. This is not necessarily optimal. As we discussed in the original paper, we can either compress the distance along the contrast direction or stretch the distance along the anatomy direction to reduce the anatomical shift. We did the former approach. But to further stretch the distance along the anatomy direction, we can change the segmentation label maps such that the label number differences between adjacent retinal layers become larger, which effectively stretches the distance along the anatomy direction. While we have not explored these alternative learned metrics, we instead provided a discussion about the potential alternative learned metrics in the discussion.

---

> ### Comment · Reviewer_ZM1d · 2025-08-03
> **Remaining Concerns After Rebuttal**
>
> Point 2:
> I acknowledge that it would be difficult to register 2 OCT volumes from two devices. However, without any ground trurh comparison, I feel like it is still impossible to evaluate whether the generated images are reliable or authentic and whether it will introduce any potential bais if you use the generated data as the dataset to train any other deep learning models.
> Point 4:
> The authors frame LMSB as a general methodological contribution. If the work is truly a method innovation, a minimum standard is to demonstrate domain-agnostic behavior on natural-image dataset or non-OCT medical modality (MRI or CT).
>
> Conversely, if the contribution is meant to be OCT-specific, then the paper must establish clear clinical value—for example, by improving RNFL/GCL thickness agreement, enhancing glaucoma classification AUC, or reducing inter-device bias in longitudinal trials. In its current state the manuscript occupies an uncomfortable middle ground: it imports a slightly modified Schrödinger-Bridge pipeline from computer vision, tests it only on OCT, yet stops short of showing any tangible ophthalmic benefit. Without either (i) multi-domain validation or (ii) concrete clinical impact, the work risks being perceived as a port-and-tweak exercise rather than a substantive advance.

---

> ### Author Response · Authors · 2025-08-04
>
> We agree with the reviewer for the remaining concerns in point 2 and point 4, and we try to address them by evaluating the proposed method on a paired OCT dataset.
>
> We have managed to identify 115 pairs of Cirrus and Spectralis volumes collected from 59 subjects and 115 eyes, which is separate from the training and testing dataset in the original paper. These 115 pairs of scans were completed contemporaneously. The retinal layer thickness measurements from these two devices disagree in general, which is an important clinical issue in OCT imaging when comparing the measurements from different devices on monitoring a subject that is scanned on different machines during the time course of the study. (This bias is the thickness measurements as reported by the scanning device.)
>
> To evaluate the impact of the proposed method on this inter-device bias, we conducted the following experiments on the paired OCT dataset: 1) Spectralis to Cirrus harmonization; and 2) Cirrus to Spectralis harmonization.
>
> In the first experiment, we harmonize the 115 Spectralis volumes to the Cirrus scanner. We computed the retinal layer thickness measurement differences between synthetic Cirrus (synC) volumes and corresponding original Cirrus (orgC) volumes, and we compared them with the differences between original Spectralis (orgS) volumes and original Cirrus (orgC) volumes. The retinal layer thickness measurements were done by first applying a deep learning segmentation algorithm and then computing the average layer thickness in a 5x5 mm2 square centered on the fovea. The results are shown in the following table, with the mean (and standard deviation) of the signed difference in units of microns (um).
>
> | | RNFL | GCIPL | INL | OPL | ONL | IS | OS | RPE |
> |:-:|:-:|:-:|:-:|:-:|:-:|:-:|:-:|:-:|
> | orgS - orgC | 9.66 (2.70)     | -4.75 (1.31)     | -2.64 (1.19)     | 0.63 (0.66)      | -4.20 (1.68)     | -0.31 (0.41)     | 1.16 (1.04)      | 1.64 (1.23)      |
> | synC - orgC | **5.44** (5.35) | **-1.89** (2.26) | **-1.79** (1.92) | **-0.16** (1.38) | **-2.33** (2.64) | **-0.06** (1.04) | **-0.18** (1.27) | **-0.27** (0.98) |
>
> The second experiment is similar to the first with the exception that we harmonize the 115 Cirrus volumes to the Spectralis scanner. We report difference in thickness measurements between the synthetic Spectralis (synS) volumes and corresponding original Spectralis (orgS) volumes, as well as the differences between original Cirrus (orgC) volumes and original Spectralis (orgS) volumes.
>
> | | RNFL | GCIPL | INL | OPL | ONL | IS | OS | RPE |
> |:-:|:-:|:-:|:-:|:-:|:-:|:-:|:-:|:-:|
> | orgC - orgS | -9.66 (2.70)     | 4.75 (1.31)     | 2.64 (1.19)     | -0.63 (0.66)    | 4.20 (1.68)     | 0.31 (0.41)     | -1.16 (1.04)    | -1.64 (1.23)    |
> | synS - orgS | **-3.23** (3.37) | **2.66** (1.92) | **1.09** (1.57) | **0.26** (0.75) | **0.48** (2.35) | **0.24** (0.48) | **0.38** (0.81) | **0.74** (1.26) |
>
> For both results, we see that harmonization helps reduce the inter-device bias for all retinal layers including the most important layer GCIPL. Moreover, paired T-tests show statistical significance (p-value < 0.05) for most cases. We need to highlight that the proposed method was originally developed such that the thickness measurements from original volumes and harmonized volumes are consistent. However, from the results in Table 1 in the original paper, we see that we still do not obtain the exact anatomy consistency with the proposed method. From these additional experiments on the paired dataset, we see that this remaining anatomy inconsistency actually helps reduce the inter-device bias.
>
> This experiment establishes a clinical value that the proposed method reduces the inter-device bias, which is one example in the reviewer's comment in point 4. However, this only partially addresses the concern in point 2 because it is an evaluation on paired data but it is still not a direct pixel-wise evaluation. But it is currently the best evaluation we can do on paired data due to the difficulty of the OCT volume registration, which the reviewer also agrees with.
>
> We hope that this addition experiment will address the reviewer's concern, and we are happy to incorporate more evaluations if the reviewer has additional concerns.

---

> > ### Comment · Reviewer_ZM1d · 2025-08-04
> > **Response**
> >
> > Thank you and you've addressed my concerns!

---

> > > ### Author Response · Authors · 2025-08-04
> > >
> > > Thank you! Your comments are really helpful!

---

### Decision · Program_Chairs · 2025-09-17

**Decision:**

Accept (poster)

**Comment:**

(a) Scientific Claims and Findings:
This paper addresses the anatomy-shift problem in medical image harmonization, which arises when using Schrödinger Bridge (SB) or entropy-regularized optimal transport directly on images. The authors propose a novel anatomy-guided latent metric Schrödinger Bridge (LMSB), which introduces an invertible neural network to map optical coherence tomography (OCT) images into a latent Euclidean space guided by anatomical segmentation maps. Key findings: LMSB reduces anatomical distortions while harmonizing OCT images across devices. It achieves superior anatomical consistency compared to DDIB and DSBM baselines. Quantitative and qualitative results demonstrate improved fidelity in preserving anatomical structures. The method is theoretically generalizable to other modalities such as MRI and ultrasound.

(b) Strengths:
1. Novelty and significance: Introduces the concept of an anatomy-guided latent metric in SB for harmonization, directly tackling a long-standing weakness of prior approaches.
2. Clear motivation: Identifies and corrects the flawed assumption in standard OT/SB (Euclidean closeness of same-anatomy images), making the contribution well-founded.
3. Methodological rigor: Strong combination of theory, algorithmic innovation, and empirical validation.
4. Comprehensive evaluation: Includes theoretical discussion, quantitative experiments (MAE, LNCC, FID), and qualitative analysis on a sizeable OCT dataset.
5. Clinical relevance: Preserves anatomy critical for downstream clinical tasks, with medical experts involved in the validation process.
6. Generality: Framework is extensible beyond OCT, applicable to other medical imaging modalities.

(c) Weaknesses and Missing Elements:
1. Limited modality scope: Demonstrations are restricted to retinal OCT; broader empirical validation (e.g., MRI, ultrasound) would strengthen claims.
2. Metric limitations: Anatomical metrics (MAE, LNCC) dominate evaluation; while FID was added in rebuttal, additional perceptual or task-driven metrics would be useful.
3. Preprocessing reliance: Some dependence on resizing/cropping steps, which may be more complex in other modalities such as ultrasound.
4. Anatomy label map encoding: Current simple numbering scheme for layers may not optimally capture inter-layer relationships.
5. Paired validation: Registration-based paired scan evaluation (e.g., PSNR/SSIM) is missing due to technical challenges.

(d) Reasons for Acceptance
The paper makes a clear conceptual advance: introducing anatomy-guided latent metrics into Schrödinger Bridge models to mitigate anatomical shifts. It provides convincing empirical results on a substantial OCT dataset, with statistical validation (t-tests) and complementary metrics (FID) confirming robustness. The work has high potential impact in clinical imaging, where anatomical fidelity is crucial. The methodology is general and extensible, making it relevant beyond the specific OCT case. The rebuttal addressed all reviewer concerns thoroughly, strengthening confidence in the results and claims.

Overall, this is a well-rounded contribution that advances both theory (metric design in OT/SB) and practice (harmonization for medical imaging).

(e) Rebuttal Period and Discussion

Reviewers raised several points:
1. Statistical validity of results: Authors added one-sided paired t-tests, confirming statistical significance.
2. Generality to other modalities: Authors argued LMSB is theoretically generalizable; challenges in preprocessing were acknowledged and discussed.
3. Ultrasound variability: Clarified as a preprocessing issue, not a limitation of LMSB itself.
4. Anatomical complexity encoding: Authors acknowledged current label encoding is suboptimal and promised elaboration.
5. Computational resources: Authors provided detailed GPU usage and clarified feasibility on a single GPU.
6. Assumptions of OT/SB: Authors reiterated why Euclidean distances are problematic and how LMSB corrects this.
7. Evaluation metrics: Added FID to measure domain consistency; results show LMSB competitive with baselines while preserving anatomy better.
8. Dataset splitting and contamination concerns: Authors clarified subject-level train/test splits with no leakage.
9. Preprocessing details: Authors provided full crop/resize pipeline and original image resolutions.
10.Expert validation: Noted two medical doctors contributed throughout development.
All concerns were addressed convincingly, and reviewers unanimously agreed on acceptance after rebuttal.

The paper introduces a principled and impactful method with clear clinical and methodological contributions.